# One mother for two species via obligate cross-species cloning in ants

Y. Juvé[1,12], C. Lutrat[1,12], A. Ha[1,12], A. Weyna[1,2], E. Lauroua[1], A. C. Afonso Silva[3], C. Roux[3], E. Schifani[4,5], C. Galkowski[6], C. Lebas[13], R. Allio[7], I. Stoyanov[8], N. Galtier[1], B. C. Schlick-Steiner[9], F. M. Steiner[9], D. Baas[10], B. Kaufmann[11] & J. Romiguier[1✉]

Living organisms are assumed to produce same-species offspring[1,2]. Here, we report a shift from this norm in *Messor ibericus*, an ant that lays individuals from two distinct species. In this life cycle, females must clone males of another species because they require their sperm to produce the worker caste. As a result, males from the same mother exhibit distinct genomes and morphologies, as they belong to species that diverged over 5 million years ago. The evolutionary history of this system appears as sexual parasitism[3] that evolved into a natural case of cross-species cloning[4,5], resulting in the maintenance of a male-only lineage cloned through distinct species' ova. We term females exhibiting this reproductive mode as xenoparous, meaning they give birth to other species as part of their life cycle.

Although clonality is the most straightforward mode of reproduction, most animal species take a more complex route[6]. In sexual species, for instance, reproduction requires the interaction of males and females, which typically means that two different morphs have to be produced[7]. Such complexity is further amplified in some species, in which females produce distinct morphs depending on seasonal conditions, population density or social caste[8–11]. Even in these extreme cases, a seemingly universal constraint persists: regardless of their morphological variation, phenotypes produced by a female invariably belong to the same species. Here, we report that this rule has been transgressed by *Messor ibericus* ants, with females producing individuals from two different species.

Previous studies on *Messor* genus ants have reported conflicting results, suggesting widespread hybridizations between species that rarely co-occur in Europe[12,13]. Here, a combination of field work, population genomic analyses and laboratory experiments provide the resolution of this paradox: females of one of the species (*M. ibericus*) clone males of the other (*Messor structor*), as they need their sperm to produce the worker caste. We discuss the evolutionary history of this natural case of cross-species cloning, which suggests a domestication-like process for exploiting another species' gametes.

## Queens depend on another species' sperm

Population genetic analyses revealed that *M. ibericus* queens are unable to produce workers without mating with males of another species. To reach this conclusion, we analysed genome-wide data in 390 individuals (Supplementary Table 1) from five European species of the *Messor* genus (phylogenetic tree in Fig. 1a and Extended Data Figs. 1 and 2). In ants, workers and queens of the same species are diploid individuals expected to be genetically similar[14]. Our data showed that this is not

the case in one out of the five species analysed. In *M. ibericus*, all worker genomes ($n = 164$) featured a 15 times higher heterozygosity than their queens or queens and workers of the four other species ($n = 127$; average of 0.797 versus 0.047 on 43,084 polymorphic sites, two-sided Wilcoxon rank-sum test $P < 2.2 \times 10^{-16}$; Fig. 1a). Such high heterozygosity levels suggest that *M. ibericus* workers are hybrids. We confirmed this hypothesis by conducting an analysis specifically designed to detect first-generation hybrids[15], which identified all *M. ibericus* workers as such (Methods and Supplementary Table 1). With the exception of one *Messor ponticus* worker, queens and individuals of the other four species were identified as non-hybrids (Supplementary Table 1).

To identify the maternal origin of hybrid workers, we conducted a phylogenetic analysis on the maternally inherited mitochondrial genome. The resulting tree suggests an *M. ibericus* maternal ancestry, as all hybrid workers share the mitochondrial genome of *M. ibericus* sexual individuals (Extended Data Fig. 2). To identify the paternal species, we conducted a phylogenetic analysis of nuclear DNA after separating the maternal and paternal alleles of the hybrid genomes (Methods). The resulting phylogenetic tree showed that hybrid workers have an *M. structor* paternal ancestry, as all paternal alleles ($n = 164$) formed a well-supported clade with individuals of this species (Extended Data Fig. 3). Finally, a population structure analysis[16] on 5,856 genes (44,191 variants) revealed that workers in *M. ibericus* colonies had virtually equal population ancestry proportions from *M. ibericus* and *M. structor* (averaging 0.49 and 0.51, respectively; Fig. 1 and Supplementary Table 1), which confirms further that they are first-generation hybrids.

These results imply that *M. ibericus* depends on hybridization for worker production, as already observed in cases of sperm parasitism[17], in which queens exploit sperm from another lineage or species to produce workers[12,18–21]. Here, *M. ibericus* queens strictly depend on males of *M. structor*, which is a well-differentiated, non-sister

[1]ISEM, University of Montpellier, CNRS, IRD, Montpellier, France. [2]Department of Ecology and Evolution, University of Lausanne, Lausanne, Switzerland. [3]Evo-Eco-Paleo UMR 8198, University of Lille, CNRS, Lille, France. [4]Department of Chemistry, Life Sciences, and Environmental Sustainability, University of Parma, Parma, Italy. [5]Institut de Biologia Evolutiva, CSIC, University of Pompeu Fabra, Barcelona, Spain. [6]Antarea, Saint-Aubin-de-Médoc, France. [7]Centre de Biologie pour la Gestion des Populations, INRAE, CIRAD, IRD, Montpellier SupAgro, Université de Montpellier, Montpellier, France. [8]Department of Developmental Biology, University of Plovdiv 'Paisii Hilendarski', Plovdiv, Bulgaria. [9]Department of Ecology, Universität Innsbruck, Innsbruck, Austria. [10]Institut NeuroMyoGène, CNRS UMR-5284, INSERM U-1314, MeLis, Université Lyon, Université Claude Bernard Lyon 1, Lyon, France. [11]LEHNA UMR 5023, Université Claude Bernard Lyon 1, CNRS, ENTPE, Villeurbanne, France. [12]These authors contributed equally: Y. Juvé, C. Lutrat, A. Ha. [13]Unaffiliated: C. Lebas. ✉e-mail: jonathan.romiguier@umontpellier.fr

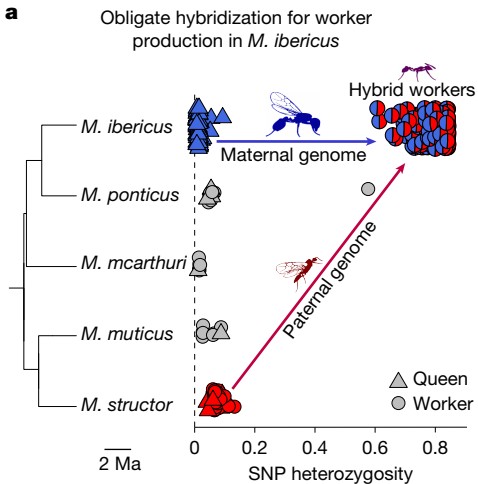

**a** Obligate hybridization for worker production in *M. ibericus*

*M. ibericus*
*M. ponticus*
*M. mcarthuri*
*M. muticus*
*M. structor*

Hybrid workers

Maternal genome

Paternal genome

△ Queen
◯ Worker

2 Ma

SNP heterozygosity
0  0.2  0.4  0.6  0.8

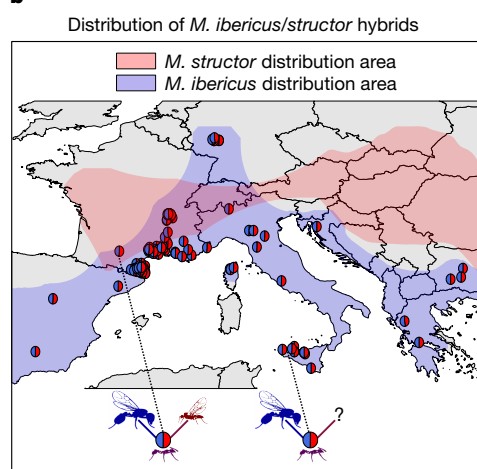

**b** Distribution of *M. ibericus/structor* hybrids

🟥 *M. structor* distribution area
🟦 *M. ibericus* distribution area

**Fig. 1 | Obligate hybridization for worker production expands beyond parental species' range. a**, Proportion of heterozygous positions on the total number of polymorphic sites (SNPs, $n = 43,084$) for queens and workers of *M. ibericus* ($n = 220$), *M. ponticus* ($n = 12$), *Messor mcarthuri* ($n = 6$), *Messor muticus* ($n = 8$) and *M. structor* ($n = 45$). Species individuals are arranged vertically according to their phylogenetic relationships (tree was built from one representative individual of each species; Extended Data Fig. 1). Each hybrid worker from *M. ibericus* colonies ($n = 164$) displays a pie chart representing its respective population ancestry proportion estimated from the fastStructure software[16], with blue and red representing, respectively, *M. ibericus* (maternal) and *M. structor* (paternal) genome proportions. Average hybrid worker

heterozygosity ($n = 164$) is significantly higher than average heterozygosity of *M. structor* queens or queens and workers of the four other species ($n = 127$; average of 0.797 versus 0.047, two-sided Wilcoxon rank-sum test, $P < 2.2 \times 10^{-16}$). **b**, Map representing the distribution of sequenced hybrid workers ($n = 164$). The distribution areas of each parental species have been estimated from our sampling and reports from the literature[13,23]. Hybrid workers localized in areas where both parental species co-occur are highlighted by a picture representing an *M. ibericus* queen (blue) with an *M. structor* male (red). Hybrid workers localized in areas without the paternal species are highlighted with the same picture but with a question mark instead of the father. SNP, single nucleotide polymorphism.

species (Fig. 1a). This finding is particularly surprising because these two species do not share the exact same distribution area[22,23]. This paradox is clearly illustrated by hybrid workers being found across Southern Europe in spite of the total absence of their paternal species (Fig. 1b; 69 Mediterranean populations with confirmed *M. ibericus* but no *M. structor* colonies found). As even more compelling evidence, first-generation hybrid workers from the Italian island of Sicily are found more than a thousand kilometres away from the closest known occurrence of their paternal species. This raises the question of how queens can hybridize in such an isolated area (Fig. 1b). To solve this conundrum, we examined males from *M. ibericus* colonies more closely.

## Queens produce males from two species

Morphological and molecular analyses showed that *M. ibericus* queens lay the *M. structor* males they require for worker production. By sampling 132 males from 26 *M. ibericus* colonies, we observed a sharp morphological dimorphism: 44% of sampled males displayed a dense pilosity (Fig. 2a), whereas the other 56% were nearly hairless (Fig. 2b). By conducting phylogenetic analyses including 62 hairy versus 24 hairless male nuclear genomes, we showed that the two morphs perfectly correspond to two different species (Extended Data Fig. 2). Whereas all hairy males group with *M. ibericus*, all hairless ones group with *M. structor*, which are two non-sister species that we estimated to have split more than 5 million years ago (Ma) (Methods, Fig. 2c and Extended Data Figs. 1 and 4). Multiple lines of evidence point to the production of males of both species by *M. ibericus* queens.

First, *M. structor* males share the same mitochondria as their *M. ibericus* nestmates, pointing to common *M. ibericus* mothers for the whole colony ($n = 24$; Fig. 2, Extended Data Fig. 2 and Supplementary Table 1). This nuclear–mitochondrial genome mismatch is unique to males found in *M. ibericus* colonies, as it has not been observed in any other *M. structor* individual when found in their own species colonies ($n = 53$; Extended Data Fig. 2 and Supplementary Table 1).

Second, genotyping 286 eggs or larvae from 5 *M. ibericus* laboratory colonies showed that 11.5% exclusively contained *M. structor* nuclear genome (Supplementary Note 1, Supplementary Table 2 and Supplementary Figs. 1 and 2). To confirm that such *M. structor* eggs were laid by *M. ibericus* queens and not workers, we isolated 16 queens and genotyped their newly produced eggs after 24 h. Again, we found that 9% of these eggs exclusively contained *M. structor* DNA (Supplementary Note 1, Supplementary Fig. 3 and Supplementary Table 3), which was not the case for broods produced by workers (see Supplementary Note 2 for details).

Third, beyond genetic evidence, direct observations confirmed the emergence of adult males of both species from a single queen colony. We monitored a laboratory colony headed by a single *M. ibericus* queen for 18 months, checking broods weekly. Among seven eggs that developed into reproductive adults, two were identified as *M. structor* (hairless) males, and three as *M. ibericus* (hairy) males. Genomic analyses confirmed their morphological identification, with their whole nuclear genome matching solely either *M. ibericus* or *M. structor* (individuals ORT3M1 to ORT5M5; Extended Data Fig. 1 and Supplementary Table 1). Despite those *M. structor* births, we confirmed that the whole genome of the mother queen solely matches *M. ibericus* (ORT3Q1; Extended Data Fig. 1 and Supplementary Table 1). Other adult male emergences of both species (one of each) have been observed in another laboratory colony after 19 months of brood monitoring (Extended Data Fig. 5 for a picture of live individuals).

Whereas male Hymenoptera typically inherit their nuclear genome from their mother through unfertilized eggs[24], our results demonstrate that *M. ibericus* queens can produce males without transmitting their nuclear genome. This observation points to androgenesis (that is, male clonality), whereby a male provides the sole source of nuclear genetic material for the embryo[25]. Embryos devoid of maternal DNA have been observed in other groups, with the fertilization of non-nucleate ovules[26] or the elimination of the maternal genome after fertilization[27]. In ants, both should spontaneously lead to males genetically identical to the sperm, as males are typically produced from haploid embryos through

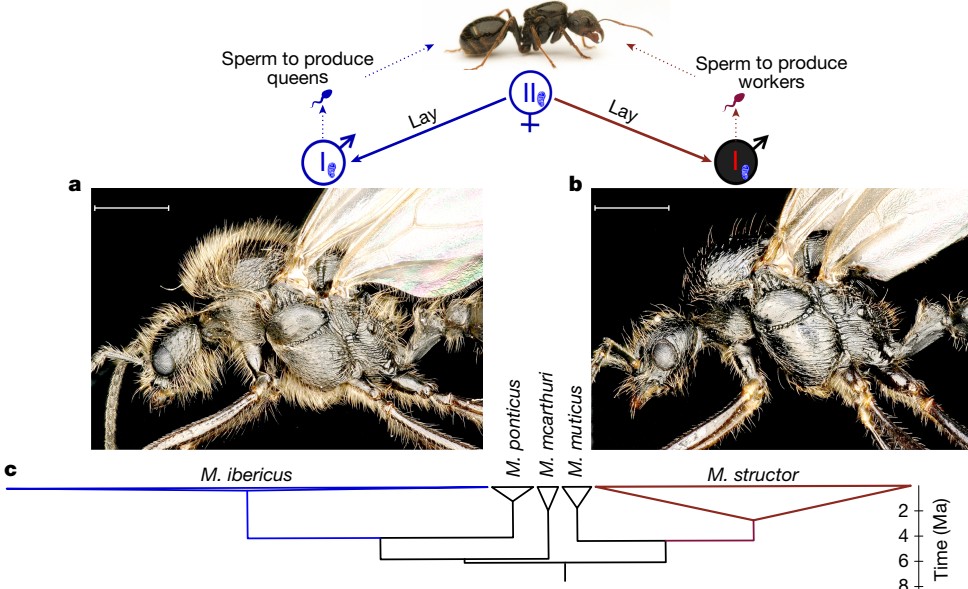

**Fig. 2 | *M. ibericus* queens lay males from two different species.** *M. ibericus* queens lay males belonging to different species that differ morphologically (symbolized by male symbols in blue and red for *M. ibericus* an *M. structor*, respectively) and genetically. *M. ibericus* and *M. structor* males produce sperm for producing either new queens or workers, respectively. All share the same mitochondria (corresponding to the *M. ibericus* mitochondria, depicted here in blue; Extended Data Fig. 2). **a**, *M. ibericus* male photo (hairy). **b**, *M. structor* male photo (hairless). **c**, Phylogenetic tree of 223 non-hybrid individuals. Based on 5,656 nuclear genes (2,780,573 bp) and simplified from Extended Data Fig. 1. All represented nodes have maximal bootstrap support (100). Triangle widths are relative to the number of individuals. Branch lengths are relative to divergence time estimated from Fig. 1 and Extended Data Fig. 4 (see Methods for details). Scale bars, 1 mm. Credit: The top picture of an ant is adapted with permission from a photo from Flickr (https://www.flickr.com) taken by M. Kukla. bp, base pairs.

haplodiploidy[24]. At the intraspecific level, several cases of ants cloning males from their own species' sperm have been observed[28–31]. Here, our results imply that this phenomenon has crossed species barriers, with male cloning from allospecific sperm stored in the spermatheca. Consistent with this explanation, *M. ibericus* queens are polyandrous and mate with both species' males, as we retrieved sperm of both *M. ibericus* and *M. structor* when sequencing the spermatheca content of a queen that gave birth to both species (ORT3QS1 in Supplementary Table 1 and Extended Data Fig. 3; see also the BAN1QS spermatheca, which again contains spermatozoa of both species).

## Maintenance of a clonal lineage of males

The combination of obligate hybridization for worker production (Fig. 1) and cross-species cloning (Fig. 2) points to the following scenario: *M. ibericus* queens first stored sperm from another species, then began to clone males from this sperm. This pathway is consistent with the widespread observation of facultative or obligate sperm parasitism[17], a well-described phenomenon in which queens use sperm from a co-occurring lineage or species to produce their workers[15,18–21,28–30,32]. This strategy may have been selected either to benefit from potential worker hybrid vigour[17] or to prevent queen-only production due to the fixation of a caste-biasing genotype[18,32]. In the ancestral state of this scenario, *M. ibericus* exploits sperm from co-occurring *M. structor* colonies (Fig. 3a), as has been observed in other *Messor* species[12,33]. In the derived state, *M. ibericus* queens directly produce the species they depend on, resulting in a clonal lineage of *M. structor* males they maintain in their colonies (Fig. 3b).

To confirm the advent of such a clonal lineage of males, we examined the two primary subdivisions of the *M. structor* nuclear phylogeny (Fig. 3c). As expected, one subdivision corresponds to a clonal lineage, consisting exclusively of nearly identical *M. structor* males, all found within *M. ibericus* colonies and carrying *M. ibericus* mitochondria (*n* = 24; Fig. 3b,c and Extended Data Fig. 1). By contrast, we retrieved a 'wild-type' lineage, which grouped all *M. structor* castes when found in their own species' colonies (*n* = 53; Fig. 3a,c and Extended Data Fig. 1). To further confirm our scenario, we tracked the exact parental origin of each hybrid worker (*n* = 164; Methods). Consistent with occurrences of both the ancestral and derived states (Fig. 3a,b), we found that the paternal genome can belong to either the 'wild-type' or 'clonal' lineage (Fig. 3c and Extended Data Fig. 3). Although most hybrid workers were fathered by clonal males (144 out of 164), the fact that some (20 out of 164) were fathered by wild-type males confirms the recent occurrence of our ancestral state hypothesis (Fig. 3a). Consistent with our scenario, ancestral state cases were restricted to a limited geographical area where both species still co-occur (for example, eastern France; Fig. 1b, Extended Data Fig. 3 and Supplementary Table 1). By contrast, derived state cases were widespread across Europe, as maintaining a clonal lineage of males is likely to have allowed rapid expansion of *M. ibericus* beyond the natural range of *M. structor* (for example, Mediterranean Europe; Fig. 1b). This pathway seems analogous to domestication[34], as *M. ibericus* co-opted *M. structor* males into its life cycle, maintaining them as a clonal lineage rather than exploiting them from the wild.

Supporting this view, the clonal lineage exhibited extremely low genetic diversity with high genetic load compared with the wild-type lineage (average synonymous nucleotide diversity $\pi_s$ of 0.00027 versus 0.0014, average ratio of non-synonymous to synonymous nucleotide diversity $\pi_n/\pi_s$ of 0.43 versus 0.21; Supplementary Table 4). This pattern is typically observed in clonal species[35,36], after rapid range expansions[37,38] or in domesticated lineages maintained by humans[39,40]. Interestingly, clonal males also differ morphologically: in a similar way that they differ from their *M. ibericus* nestmates (Fig. 2), they also seemed hairless compared with their wild-type counterparts (Fig. 3d,e). More generally, this clonal morph differs on several other criteria, standing out as the most divergent compared with the wild-type and *M. ibericus* males (Supplementary Note 3 for details and Supplementary Figs. 4–6), akin to the morphological divergence of domesticated species compared with their wild relatives[41]. Such a stark morphological

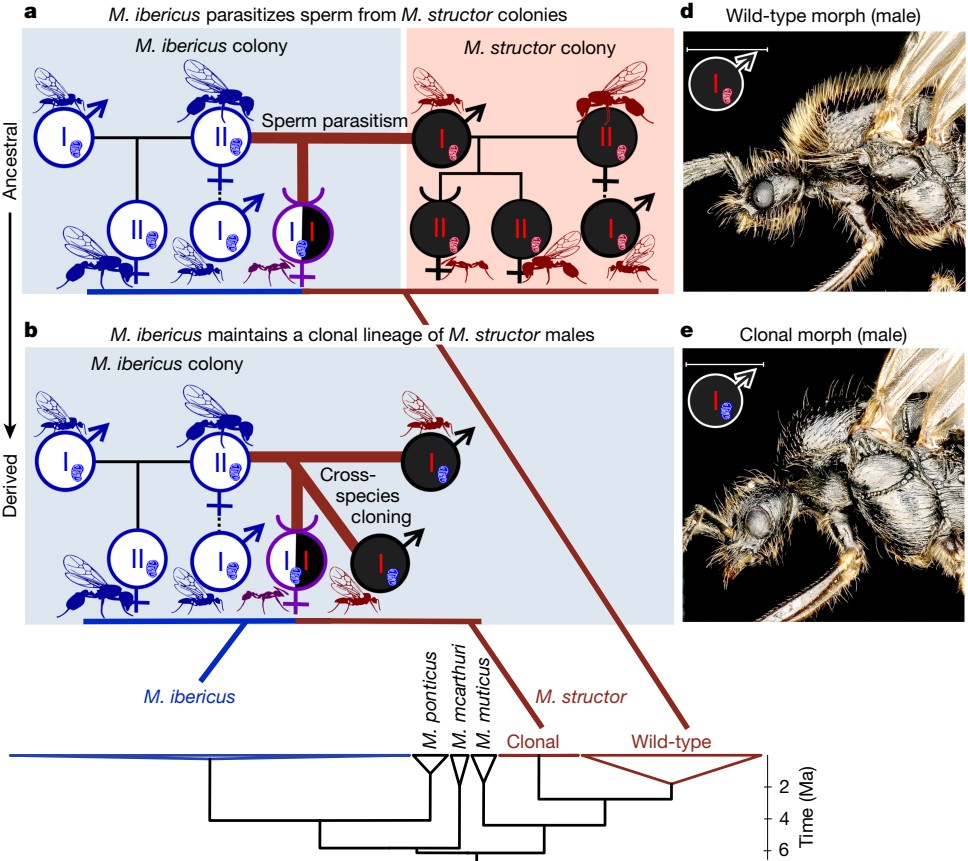

**a** *M. ibericus* parasitizes sperm from *M. structor* colonies

*M. ibericus* colony   *M. structor* colony

Sperm parasitism

**b** *M. ibericus* maintains a clonal lineage of *M. structor* males

*M. ibericus* colony

Cross-species cloning

*M. ibericus*

*M. ponticus*
*M. mcarthuri*
*M. muticus*
*M. structor*
Clonal   Wild-type

Time (Ma)
2
4
6

**c** *M. structor* is divided into 'clonal' and 'wild-type' lineages

**d** Wild-type morph (male)

**e** Clonal morph (male)

**Fig. 3 | Evolution of obligate cross-species cloning from sperm parasitism is reflected by different genetic and morphological lineages within *M. structor*.** **a**, Ancestral state of the *M. ibericus* reproductive system; $n = 20$ colonies deduced to correspond to this state have been sampled (Supplementary Table 1). **b**, Derived state of the *M. ibericus* reproductive system; $n = 130$ colonies deduced to correspond to this state have been sampled (Supplementary Table 1). Note that *M. structor* males have an *M. ibericus* mitochondrial genome, which is indicated with a red chromosome and a blue mitochondrion. **c**, Phylogenetic tree simplified from Extended Data Fig. 1 (as in Fig. 2c). Links to **a** and **b** are based on Extended Data Fig. 3, in which hybrid workers have been separated into paternal and maternal genomes. *M. structor* 'clonal' lineage stands for a clade composed of males from *M. ibericus* nests and the paternal genome of their worker daughters (derived state). *M. structor* 'wild-type' lineage stands for a clade composed of all castes from normal *M. structor* nests and the paternal genome of some hybrid workers found in *M. ibericus* co-occurring nests (ancestral state). **d**, Photo of *M. structor* males from *M. structor* colonies (hairy). **e**, Photo of *M. structor* males from *M. ibericus* colonies (hairless). Scale bars, 1 mm.

difference does not necessarily result from a selection process. Instead, this difference may have been randomly retained from ancestral polymorphism, or may be due to incompatibilities between the nuclear and mitochondrial genomes of the two species (Fig. 3b) or plasticity due to different rearing conditions when born and kept within *M. ibericus* nests.

To assess whether clonal males can escape their 'domesticated' situation by mating with their wild female counterparts, we conducted a detailed analysis on 45 *M. structor* genomes to detect potential hybrids (Supplementary Note 4). Our findings confirmed that such events are at present non-existent or extremely rare, as we did not identify any hybrid between clonal and wild-type lineages (Supplementary Fig. 7). Similarly to typical cases of domestication, this raises the question of whether recent genetic isolation from wild populations warrants a different species classification[42]. Further analyses therefore support the idea that clonal males still belong to *M. structor*, as phylogenetic conflict (Supplementary Fig. 8a), population genetic structure (Supplementary Fig. 8b), species delimitation inferences (Supplementary Fig. 8c,d), low $F_{st}$ fixation index (Supplementary Fig. 9), low genetic divergence (Supplementary Fig. 10a) and high historical gene flow (Supplementary Fig. 10b) are all consistent to support clonal and wild-type lineages as part of the same species (see Supplementary Note 4 for details).

Taken together, these results further support the idea that clonal males should be characterized as a domesticated lineage of *M. structor*. All in all, this means that *M. ibericus* females interact with up to three males that are morphologically and genetically distinct (*M. ibericus*, 'domesticated' *M. structor* and 'wild' *M. structor* males; Extended Data Fig. 6), laying two of them (Fig. 2) and mating with the three (Fig. 3).

## Discussion

To our knowledge, females needing to clone members of another species have not previously been observed. Although cross-species cloning has been reported in hermaphrodite conifers and clams[25], these are instances of male parasites occasionally using other species' eggs. In such cases, producing males of another species is not in the interest of females, as they are incidental victims of parasitism. This contrasts with the system reported here, for which producing another species' male is not an accident, but a female life cycle requirement. We suggest defining such females as xenoparous, meaning they need to produce individuals of another species as part of their life cycle. This shows the evolution of xenoparity (xeno-, meaning 'foreign, strange, different', and -parity, meaning 'produce, bring forth, give birth'), which is the need to propagate another species' genome by means of its own eggs.

Transition towards xenoparity seems to result from sexual evolution along a parasitism–mutualism continuum. Similar to several other harvester ant species, *M. ibericus* first transitioned into obligate sperm parasitism[12,17] (Fig. 3a), a situation in which they lost the ability to produce workers by themselves due to epistatic incompatibilities[18,43] or selfish caste-biasing genotypes[32]. Although not the most straightforward path towards xenoparity, this situation might have evolved towards reciprocal sperm parasitism, a form of sperm mutualism seen in other harvester ants in which two lineages depend on each other's sperm for worker production[12,18,21]. Whether it be in the case of simple or reciprocal parasitism, dependence on males from another species is sub-optimal for queens, as it requires them to mate with two different male partners and restricts their colonies to the geographic range of their host. By producing the required species' males in their own colonies (Fig. 3b), *M. ibericus* has gained a clear advantage, as it maintains obligate hybridization while minimizing the inherent constraints (Extended Data Fig. 7). Investigating the male cloning mechanism will help to determine whether this developmental innovation is analogous to male parasitism[25] or unique to the *M. ibericus* reproductive system.

While trapped in the life cycle of a species exploiting their sperm, clonal males propagate their genome through the reproductive efforts and parental care of *M. ibericus*. In a sense, clonal males can be viewed as a perfected form of male parasites, as they are essential to their female hosts but reproduce at the expense of their ova. By depending on each other's gametes, both species have intertwined their life cycles, evolving from sexual parasitism[3] to sexual co-dependency (Extended Data Fig. 8). In spite of this, females seem to control the terms of the relationship, as our data on brood genotyping suggest that they impose the timing of male eggs' development and maturity (Supplementary Note 1). Such a situation seems akin to a sexual domestication, as *M. ibericus* controls the reproduction of a species it first exploited from the wild.

Although matching all criteria of domestication[34], the relationship we describe is both more intimate and integrated than the most remarkable examples known so far, from human-driven domestication[40] to lichen symbiosis[44]. Contrary to such examples, both partners are obligate mating partners, as the domesticating species is directly cloning the domesticated one by means of its own egg cytoplasms. Such replication of an alien genome within one's own cytoplasm echoes the endosymbiotic domestication of organelles (for example, mitochondria) within eukaryotic cells[45,46]. Clonal males may thus be regarded as organelles at the superorganism level[47,48], resulting from the integration of this alien genome into a colony that directly replicates it. This leads to colonies producing the greatest diversity of individuals, differing in terms of sexes, castes and species, each with a dedicated role within a cohesive reproductive unit. Besides revealing a reproductive mode under which one species needs to clone another, such a 'two-species superorganism' challenges the usual boundaries of individuality. Major evolutionary transition in individuality occurs when distinct entities evolve into an integrated, higher-level unit[49–51]. As two species have become sexually interdependent in such an integrated entity, evolution towards xenoparity exemplifies how such transitions can occur through a sexual domestication process.

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

## Methods

### Sampling

To better understand hybridization patterns between *M. ibericus* and *M. structor*, we gradually sampled individuals of both species, along with their respective closest relative species across Europe (*M. ponticus*, *M. muticus* and *M. mcarthuri*)[13]. In total, we sequenced 377 individuals from 125 different populations (280 *M. ibericus*, 8 *M. ponticus*, 6 *M. mcarthuri*, 75 *M. structor* and 8 *M. muticus*; Supplementary Table 1). From these individuals, we sequenced one reference genome of *M. ibericus* with long-read sequencing, 327 genomes with short-read sequencing and 51 transcriptomes. The previously published transcriptomes of seven *M. ibericus*, one *M. structor* and five *M. ponticus*[12] were added to the final dataset. Short-read sequencing of a *Messor wasmanni* worker was also added and subsequently used as an outgroup. We also dissected the spermatheca of two *M. ibericus* queens and sequenced their contents by short-read sequencing. We kept and monitored 65 colonies of *M. ibericus* in artificial nests. Colonies were kept in a room at 25 °C and 40% humidity and were fed with grass seeds.

### Reference genome assembly

To obtain a reference genome for our population genomic dataset, high-molecular-weight DNA extraction of an *M. ibericus* queen from the Passa population was sequenced using PacBio long-read sequencing[52]. Illumina short-read sequencing from the same individual was also produced to polish the genome assembly. The genome assembly was performed using the Wtdbg2 assembler v.2.5 (ref. 53). The PacBio Sequel reads were processed assuming a genome size of around 300 megabases and using preset2 settings which are more appropriate for genome sizes lower than 1 gigabase. The initial assembly was then polished using POLCA[54], a tool incorporated in the MaSuRCA assembler (v.3.4.1), leveraging its ability to correct discrepancies using Illumina short reads. The process was enhanced further by using Next-Polish v.1.3.1 (ref. 55) for error correction, using both the short and long reads for correction with default settings. To improve the assembly's contiguity, we used RagTag v.1.0.2 with the scaffold command[56], using a *Messor capitatus* genome (GAGA-0413_Messor_capitatus.fasta)[57] as a reference. Following the scaffolding, we applied another round of polishing with POLCA before proceeding to fill the gaps in the assembly with TGS-GapCloser (v.1.1.1)[58]. This program uses raw long reads and Racon[59] to polish the filled gaps. Finally, this was followed up with a final round of polishing with POLCA and NextPolish to ensure the accuracy of the assembly.

The resulting assembly was evaluated using QUAST (v.5.0)[60], with a total assembly length of 310,325,892 bp divided in 618 contigs, GC% of 36.82 and N50 of 12,028,351 bp. We then ran a BUSCO (v.4.0.5)[61] analysis to evaluate the completeness of the genome. We used default parameters, database hymenoptera_odb10 and lineage dataset *Camponotus floridanus*. We retrieved complete sequences of 97.7% of the 5,991 Hymenoptera single-copy orthologues.

### DNA and RNA extraction and library preparation

To sequence a population genomic dataset for each species, we performed Illumina short-read sequencing of either whole genomes or transcriptomes on 390 samples. DNA extractions were performed using Macherey-Nagel NucleoMag Tissue kit with an extra RNase step. Library preparation for whole-genome sequencing was performed using customized Illumina protocols[62,63].

For RNA extraction, we killed individuals in liquid nitrogen and conserved them at −80 °C. RNA extractions were performed using the following protocol. First, the sample was homogenized with ceramic beads in 1 ml of TRIzol solution (3 × 30 min at 6.5 °C). The homogenized samples were incubated in TRIzol for 5 min at room temperature. Next, 200 µl of chloroform was added and the mixture was vortexed vigorously for 15 s, followed by incubation for 5 min at room temperature.

The mixture was centrifuged for 25 min at 12,000 rpm (4 °C). The upper aqueous layer (~500 µl) was transferred to a new tube. Next, 1 µl of glycogen blue (RNAse-free, Invitrogen, 15 mg ml$^{-1}$, catalogue no. AM9516) was added. The mixture was then vortexed and incubated overnight at −20 °C. The sample was centrifuged for 30 min at 12,000 rpm (4 °C). After centrifugation, the supernatant was discarded and 1 ml of 80% EtOH was added. The mixture was vortexed briefly and centrifuged for 5 min at 12,000 rpm (4 °C). All supernatants were removed and the pellet was allowed to air dry for 15–20 min at room temperature. Library preparations for transcriptome sequencing were made using the Roche KAPA mRNA HyperPrep Kit (catalogue no. 08098115702).

### Read mapping

To map the short-read sequencing data onto the reference genome, Illumina reads of the 390 samples (326 whole-genome sequencing, 64 RNA sequencing) were trimmed and filtered using fastp v.0.23.2 (ref. 64), requiring a minimum quality score of 20 (-q 20), discarding reads with more than 70% unqualified bases (-u 70) or 40 unknown bases (-n 40) and retaining only reads longer than 40 bases after trimming (-l 40). We then mapped all the filtered reads to the reference genome of *M. ibericus* using BWA-MEM2 (v.2.2.1)[65] with default parameters. Unmapped reads and secondary alignments were discarded using SAMtools (v.1.15.1)[66] with the view command and option -F 260.

### Coding sequence search

To ensure reliable population genetic analyses, we focused on coding sequences of highly conserved single-copy orthologue genes. For this, we produced a sequence alignment map file (bam file) for single-copy orthologue genes of the nuclear genome using the following approach. From bam file alignments, we isolated the reads that overlapped coding regions of the reference genome BUSCO genes, using samtools view together with option -L and a bed file obtained from the BUSCO output. Retrieved reads were then realigned to the 5,856 BUSCO genes of the reference genome using BWA-MEM2 with default parameters. Most of the following analyses were conducted on these 5,856 genes.

To retrieve mitochondrial genes, we first identified and isolated reads of mitochondrial origin using seven mitochondrial genomes of the Myrmicinae sub-family (*Myrmica scabrinodis*, *Cardiocondyla obscurior*, *Solenopsis invicta*, *Solenopsis geminata*, *Solenopsis richteri*, *Atta laevigata* and *Wasmannia auropunctata*) as reference and mirabait (v.4.9)[67] with options -D 50 -k 31 -n 2. Corresponding mitochondrial reads were then assembled using megahit v.1.2.9 (ref. 68) with kmer size every 10 bp from 31 to 101 bp (--k-min 31 --k-max 101 --k-step 10). We analysed assemblies using mitofinder (v.1.4.2)[69] with default options to identify and retrieve mitochondrial genes.

### Variant calling

We called variants (SNPs) using GATK (v.4.3)[70]. First, we obtained precalling variant files by using gatk HaplotypeCaller with default parameters and option -ERC BP_RESOLUTION on each of 390 individual realignment files (bam files). All individuals were then pooled to produce SNP calls for the whole dataset using gatk MergeVcfs with default options. We filtered SNPs using vcftools (v.0.1.16)[71], keeping only variants with a genotype quality of more than 10 (--minGQ 10). The resulting vcf file is available from the Zenodo repository (https://zenodo.org/records/11506545 (ref. 72)).

### Heterozygosity and hybrid detection

To detect hybrids among sequenced individuals, we computed SNP heterozygosity. For this, we filtered further the SNPs using vcftools (--remove-indels --maf 0.05 --max-missing 0.8) before computing the number of heterozygous sites for each individual using a Python home-made script (available from the Zenodo repository via https://zenodo.org/records/11506545 (ref. 72)). Number of heterozygous positions per individual was then divided by the total number of

polymorphic sites ($n$ = 43,084; Supplementary Table 1). We expected to observe higher heterozygosity values in hybrid individuals. Our analysis clearly confirmed this pattern for *M. ibericus* workers compared with other individuals (two-sided Wilcoxon rank-sum test, $P < 2.2 \times 10^{-16}$; Fig. 1a). A single *M. ponticus* worker showed similar heterozygosity values (0.58) and has been retained with *M. ibericus* workers to be tested for hybrid status below.

To further confirm the hybrid status of *M. ibericus* workers, we used a Bayesian approach designed to specifically detect first-generation hybrids[15]. To reduce computing time and avoid discrepancies between whole-genome and RNA sequencing data, we restricted the analysis to 833 highly expressed housekeeping genes, that is, keeping only universal genes common to metazoa found in the OrthoDB[73] dataset metazoa_odb10. The approach estimated the $\gamma$ parameter, which is a measure of the heterozygosity acquired during the divergence of two separated populations. We expected the $\gamma$ hybrid index to be higher in hybrid genomes (164 *M. ibericus* workers and 1 *M. ponticus*) and greater than zero in hybrid genomes (all other individuals). We found a clear non-overlapping distribution of higher $\gamma$ hybrid index in *M. ibericus* workers and one hybrid *M. ponticus* worker (average of 0.00186, range from 0.0002 to 0.0024) versus all other individuals which have very close to zero $\gamma$ values (average of $1.86 \times 10^{-6}$, range from $6.598 \times 10^{-7}$ to $6.461 \times 10^{-6}$), with a highly significant difference (two-sided Wilcoxon rank-sum test, $P < 2.2 \times 10^{-16}$).

### Nuclear phylogeny inference

Because hybrid individuals contain genomes from different species, they can bias the inference of phylogenetic relationships. To obtain clear relationships among *Messor* species, we thus first excluded hybrid individuals and built a phylogeny on the basis of nuclear genes of the 223 non-hybrid individuals. Individual variant calling files (vcf files) were treated separately from this point on. Indels were removed from each individual vcf file using vcftools (--remove-indels). Consensus sequences for 5,856 single-copy orthologue genes (BUSCO genes) were extracted from vcf files using bcftools (v.1.15.1) consensus[74], with heterozygous position treated as missing data. Positions with depth coverage of less than 3 were replaced by a gap and genes with more than 50% gaps were excluded. Alignments for each of 5,856 single-copy genes were built separately using MAFFT[75]. Outgroups were added using the macse alignTwoProfiles command[76]: *M. wasmanni* (this study), *Messor barbarus*[77], *Aphaenogaster floridana*[78] and *Acromyrmex echinatior*[79]. We concatenated the 5,856 gene alignments then trimmed the obtained supermatrix using trimal by removing sites with more than 5% missing data, resulting in an alignment of 2,780,573 sites. We inferred a phylogenetic tree using IQ-TREE (v.2.07)[80] with a GTR + I + F + G4 model (general time reversible model with proportion of invariant sites, empirical base frequencies and a gamma distribution with four rate categories) and 1,000 ultrafast bootstraps (-bb 1000). The topology of the resulting tree is available in Extended Data Fig. 1. By removing hybrid individuals, we expected clear parental relationships among species. As expected, all nodes defining the species relationships exhibited a maximal bootstrap support of 100.

### Mitochondrial phylogeny inference

To identify the maternal species of hybrid individuals, we built a mitochondrial phylogeny including our 390 individuals (Supplementary Table 1), and we first aligned separately the 15 mitochondrial genes using MAFFT (v.7.490)[75]. We added data of the *M. wasmanni* genome as an outgroup. We then concatenated the alignments before cleaning the resulting supermatrix using trimal (v.1.4)[81] with the automated1 option. The resulting 2,585-site supermatrix was then used to infer a phylogenetic tree with IQ-TREE (v.2.07)[80], using the MFP option for automatically selecting the substitution model with 1,000 ultrafast bootstraps (-bb 1000). We expected that all individuals sampled from *M. ibericus* colonies—including *M. ibericus* males and females, hybrid workers and *M. structor* males—were laid by *M. ibericus* queens. Given that the mitochondrial genome is maternally inherited, we expected these individuals to group within the same clade. As expected, all individuals from *M. ibericus* colonies grouped in the same clade, regardless of their hybrid status or nuclear genome species origin (Extended Data Fig. 2).

### Divergence time estimation

We estimated the divergence times of our species with MCMCtree from the PAML package (v.4.10.7)[82]. For this, we first built a phylogenetic tree with one individual per species, as recommended by the MCMCtree manual. Representative individuals were selected on the basis of their coverage (Supplementary Table 1): Y15452-1 for *M. muticus*, Y16370-1 for *M. mcarthuri* and Y14753-1 for *M. ponticus*, Y15268-1 for *M. structor* (wild-type lineage), SH19-04 for *M. structor* (clonal lineage) and the long-read reference genome for *M. ibericus*. The same outgroups as for the previous analysis were kept. We trimmed the supermatrix by removing all sites with at least one gap, resulting in an alignment of 6,089,069 sites. We inferred a phylogenetic tree using IQ-TREE (v.2.07)[80] with a GTR + I + F + G4 model and 1,000 ultrafast bootstraps (-bb 1000). The resulting tree had similar species relationships as the previous one. All nodes had a maximal bootstrap support of 100. We used this tree to constrain the topology of the divergence time estimation. This analysis was run using the same 6,089,069 supermatrix by using MCMCtree rapid approximate likelihood computation[83]. Based on a time-calibrated phylogeny of the Stenammini tribe[84], we constrained the root node with soft bounds from −71.2 to −101.8 Ma, corresponding to the lower and upper bound of the 95% highest posterior density of that study's main analysis. Similarly, we set soft bounds on the common ancestor of *M. barbarus* and *M. wasmanni* from −6.4 to −12.5 Ma and the common ancestor of *A. floridana* and all *Messor* from −12.7 to −21.1 Ma. We ran two runs of the analysis with the correlated rates model, HKY85 substitution model for 600 million generations. We confirmed convergence and sufficient effective sample sizes (≫200) for all parameters using Tracer v.1.7.2 (ref. 85). The resulting tree with confidence intervals of estimated divergence time is available in Extended Data Fig. 4.

We used the average divergence times of this tree's nodes as secondary constraints for the mitochondrial and nuclear trees with all individuals, using the least squares method in IQ-TREE (v.2.12)[80] with the −date option to obtain an ultrametric tree as used for Figs. 2 and 3 and Extended Data Figs. 1–3.

### Phasing maternal and paternal alleles of hybrid individuals

To identify the parental species of each hybrid individual, we developed a custom phasing approach for separating paternal and maternal alleles of each hybrid individual. SNP calls were first isolated after filtering out indels (vcftools −remove-indels). Previous results indicated that *M. ibericus* is the maternal species of all hybrid workers, as hybrids are found only in *M. ibericus* queen colonies (see also Fig. 1 and Extended Data Fig. 1). Consequently, alleles of hybrid individuals that do not match *M. ibericus* are expected to belong to the paternal species. Given the exceptionally low genetic diversity of *M. ibericus* ($\pi_s$ of 0.00045; Supplementary Table 4), the risk of confusing intraspecific polymorphism with paternal alleles is minimized. We exploited this specificity by writing a Python script comparing each SNP of each hybrid worker with variants of a reference maternal genome (*M. ibericus* queen genome with the highest coverage, SH19-06). The script parses a vcf file position by position, and applies the following approach:

When a site is heterozygous in the hybrid worker (for example, A/G) and one of the nucleotides matches the maternal reference at homozygous state (for example, A/A), this nucleotide is assigned as maternal (for example, A) and the other is assigned as paternal (for example, G). In all other cases (heterozygous position in maternal reference, no matching allele between hybrid and maternal references), an N is assigned to both the paternal and maternal alleles. N is also assigned when the

coverage of a site is below 3 reads in the focal hybrid or the reference maternal genome. Once the paternal and maternal nucleotides have been discriminated at all sites, maternal and paternal sequences are reconstructed using bcftools (v.1.15.1) consensus[74] (default options).

The same approach was used for: (1) spermatheca content of *M. ibericus* queen; (2) males laid by orphaned *M. ibericus*/*structor* workers; and (3) the only hybrid *M. ponticus* worker (using the *M. ponticus* queen RDNIPQ).

### Phylogenetic analysis of paternal and maternal alleles of hybrids

To identify the paternal species of hybrid individuals, we built a phylogenetic tree including non-hybrid genomes and paternal + maternal haplomes of hybrid genomes inferred from the previous analysis. The maternal and paternal sequences of the 5,856 orthologue genes were aligned with the corresponding data from non-hybrid individuals and then concatenated. We applied the same filtering as for the non-hybrid individual supermatrix (no more than 5% missing data). We inferred a phylogenetic tree from the resulting supermatrix (1,089,038 bp, 559 haplomes) using IQ-TREE (GTR + I + F + G4 model, 1,000 ultrafast bootstraps). We deduced maternal and paternal species of hybrid individuals from the phylogenetic placement of their corresponding maternal and paternal haplomes (Extended Data Fig. 3). Given their hybrid nature, we expected that the paternal haplomes of *M. ibericus* workers would group with *M. structor*. As expected, all *M. ibericus* workers (*n* = 164) were identified as hybrids, with *M. ibericus* mothers and *M. structor* fathers, supported by a maximal bootstrap value of 100 for each haplome grouping with its corresponding species of origin (Extended Data Fig. 3).

Additionally, we expected the spermatheca of *M. ibericus* queens to contain sperm from both *M. ibericus* and *M. structor* males. As expected, the sequences from the spermatheca content grouped with either the *M. ibericus* clade or the *M. structor* clade with maximal bootstrap support value of 100 (Extended Data Fig. 3). The hybrid *M. ponticus* worker was identified as a hybrid between *M. ponticus* mother and *M. structor* father with maximal bootstrap support value of 100 (Extended Data Fig. 3).

### Population structure analysis

To estimate the population ancestry proportions of hybrids, we selected *M. ibericus* and *M. structor* individuals from variant files (same variant filtering as for SNP heterozygosity computing) and then produced a bed file using PLINK[86] (v.1.90b6.21) before using fastStructure[16] (v.1.0) with *k* = 2. Because our hybrid detection approach is designed to detect first-generation hybrids, we expected each hybrid worker to exhibit population ancestry close to 50% from each parental species. As expected, we obtained average proportions of 0.49 and 0.51 for *M. ibericus* and *M. structor* ancestry, respectively. The resulting population ancestry values are detailed in Supplementary Table 1 and visualized in Fig. 1.

### Synonymous and non-synonymous polymorphism

To estimate population size and the associated genetic load of each species or lineage, we computed the synonymous polymorphism, $\pi_s$, and the ratio of non-synonymous over synonymous polymorphism, $\pi_n/\pi_s$ (ref. 87). The $\pi_n/\pi_s$ ratio estimations being very sensitive to SNP call errors, we selected only the genomes and transcriptomes with more than 15× coverage for these analyses. We used the program dNdSpiNpiS (v.1.0) available from this link (https://kimura.univ-montp2.fr/PopPhyl/index.php?section=tools), with options -allow_internal_bc=1 -compute_distances=1 -gapN_site=4 -gapN_seq=0.2. Results are available in Supplementary Table 4 with their respective confidence intervals. As expected in the case of low effective population size and high genetic load for a clonal lineage, we retrieved very low values of $\pi_s$ (0.00027, confidence interval 0.00021–0.00033) and high values of $\pi_n/\pi_s$ (0.427, confidence interval 0.378–0.485) in the *M. structor* clonal male lineage.

### PCR tests for species identification

We developed a PCR test designed to quickly identify *M. ibericus* (queens or males), *M. structor* (all castes) and hybrid individuals (*M. ibericus* workers). On the basis of our genomes, we designed a combination of two primers (namely, CL0001: CCACTGTGGCGTACCTACC; and CL0002: CTACACGTACACGCGACAC) to amplify different microsatellite fragment lengths depending on the species: a 247-bp fragment for *M. ibericus*, a 467-bp fragment for *M. structor* and both fragment lengths in hybrid individuals (*M. ibericus* workers). DNA extractions were conducted using the Phire Tissue Direct kit (Fisher), following the two-step protocol with various tissue types crunched with a pillar: eggs, partial larva, finely cut adult leg or finely cut adult wing. Amplifications were carried out in a 10-µl reaction volume comprising 5 µl of Phire MasterMix 2X (ref K0171), 3 µl of water, 0.5 µl of each primer (10 µM) and 1 µl of DNA extract. The PCR conditions were 5 min at 98 °C, followed by 35 cycles with 10 s at 98 °C, 10 s at 66 °C and 15 s at 72 °C, and a final extension of 1 min at 72 °C. PCR products were run on a 1% agarose gel, resulting in a pattern of short, long or short/long fragments depending on the species/caste combination. We validated the approach on adult individuals of various caste and colonies with previously sequenced genomes: *M. ibericus* queens (*n* = 10), *M. ibericus*/*structor* hybrid workers (*n* = 14), *M. structor* clonal males (*n* = 9), *M. structor* queens (*n* = 3) and *M. structor* workers (*n* = 8). As expected, short fragments were observed for *M. ibericus* queens, long fragments for *M. structor* (queens, workers, clonal males) and long/short fragments for *M. ibericus*/*structor* hybrid workers. Three different test runs confirmed the reliability of the approach (Supplementary Fig. 1).

### Reporting summary

Further information on research design is available in the Nature Portfolio Reporting Summary linked to this article.

## Data availability

Raw reads of genetic data are deposited at the NCBI under project ID PRJNA1145159, with SRA IDs and all data supporting the results of the study indicated for each sample in Supplementary Table 1. Reference genomes, genetic variation data and phylogenetic analyses used for producing the results of the study are available at Zenodo (https://doi.org/10.5281/zenodo.11506545)[72]. We also used the following datasets from the orthoDB database (https://www.orthodb.org/): hymenoptera_odb10 (https://busco-data.ezlab.org/v5/data/lineages/hymenoptera_odb10.2024-01-08.tar.gz) and metazoa_odb10 (https://busco-data.ezlab.org/v5/data/lineages/metazoa_odb10.2024-01-08.tar.gz).

## Code availability

Scripts used for producing the results of the study are available at Zenodo (https://doi.org/10.5281/zenodo.11506545)[72].

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

**Acknowledgements** We thank R. Blatrix, M. Menchetti, E. Genduso, A. François, J. Fusier and P. Melon for help with the sampling. We thank M. Tilak, L. Benoit, D. Dambier and C. La Mendola for help and advice with wet lab work. We thank M. Kukla, T. Holtom, M. Challe, L. Soldati and the Centre de Biologie pour la Gestion des Populations (CBGP) for help with the ant pictures. We thank M. Challe and M. Wehrung for maintenance of ant colonies. We thank S. Puechmaille, B. Nabholz, F. Delsuc, C. Scornavacca, P. David, C. Doums, R. Libbrecht, R. Blatrix, L. Després, C. Smadja, T. Leroy, E. Loire, A. Bernadou and A. Lenoir for useful discussions. This work was funded by the European Research Council grant no. ERC-2020-StG-948688 RoyalMess attributed to J.R.

**Author contributions** J.R., Y.J., B.K., E.S., I.S., C.G., B.C.S.-S., F.M.S., C. Lebas, A.W., C. Lutrat, A.H. and E.L. sampled individuals. E.L., C. Lutrat, B.K., D.B., R.A. and J.R. performed lab work. A.C.A.S. supervised by C.R. and J.R. assembled the reference genome. A.W. supervised by J.R. and N.G. performed preliminary analyses of genetic data. C. Lutrat and J.R. monitored ant colonies in artificial nests and analysed egg genotyping. A.H. and J.R. performed analyses of species delimitation. Y.J., J.R., B.C.S.-S. and F.M.S. carried out the morphological descriptions and analyses. J.R. performed the final analyses of population genetics, phylogeny and molecular dating, conducted the statistical analyses and produced all the figures. J.R. wrote the manuscript with comments from all co-authors. J.R. conceived the project, acquired the funding and supervised all experiments.

**Competing interests** The authors declare no competing interests.

**Additional information**
**Correspondence and requests for materials** should be addressed to J. Romiguier.

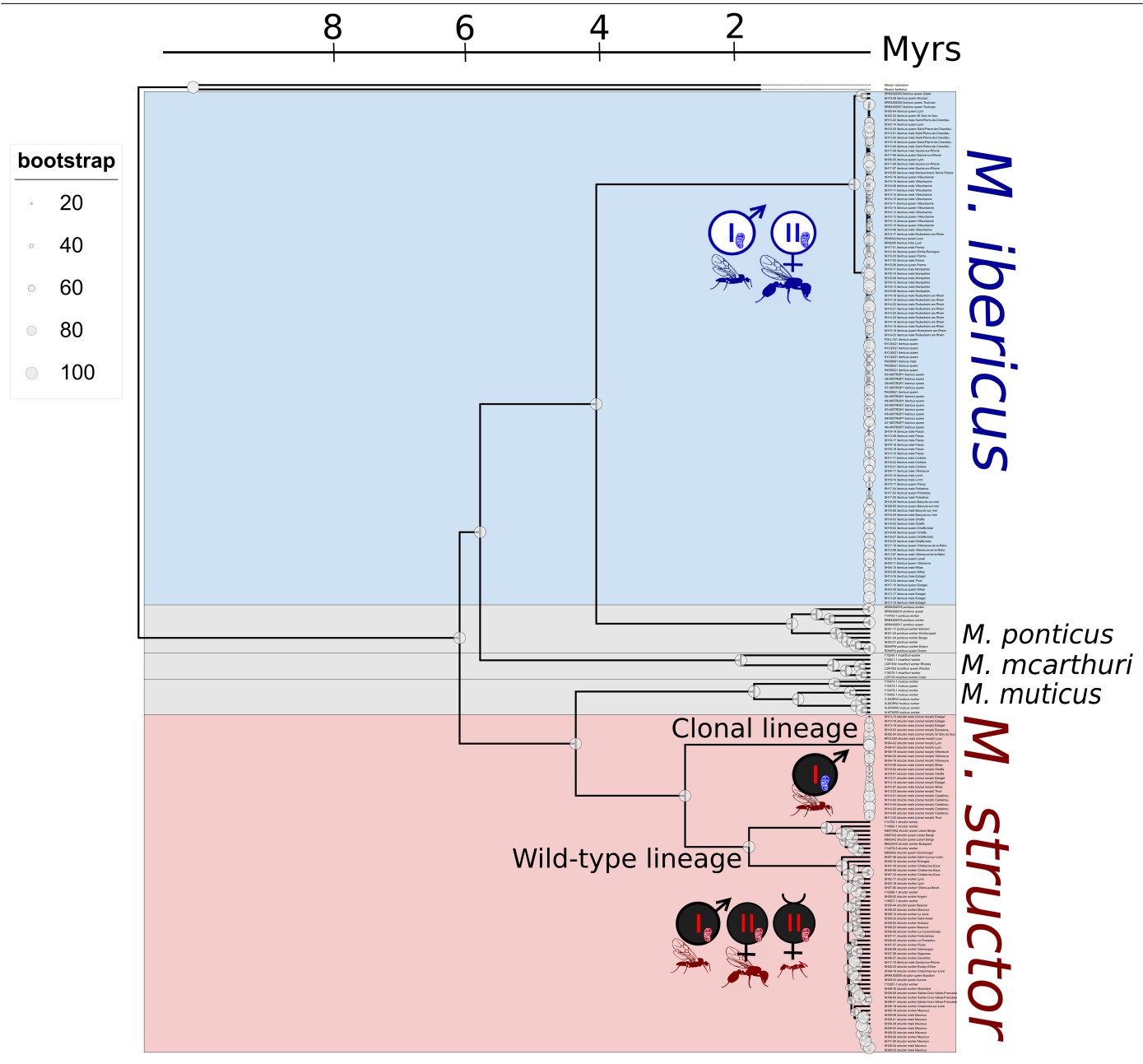

**Extended Data Fig. 1 | Molecular phylogeny of 223 non-hybrid *Messor* individuals.** The topology of the tree was produced from a supermatrix of 2,780,573 sites using *IQ-TREE* (model GTR+I+F+G4, 1,000 ultrafast bootstraps). *M. ibericus* and *M. structor* clades are illustrated with their respective castes. Red and blue bars/mitochondria indicate respectively *M. structor* and *M. ibericus* nuclear/mitochondrial genomes. All deep nodes (defining a species or a relationship among species) have maximal bootstrap support (100). Two distant outgroups of the tree (*Aphaenogaster floridana* and *Acromyrmex echinatior*) were removed from the figure for better readability.

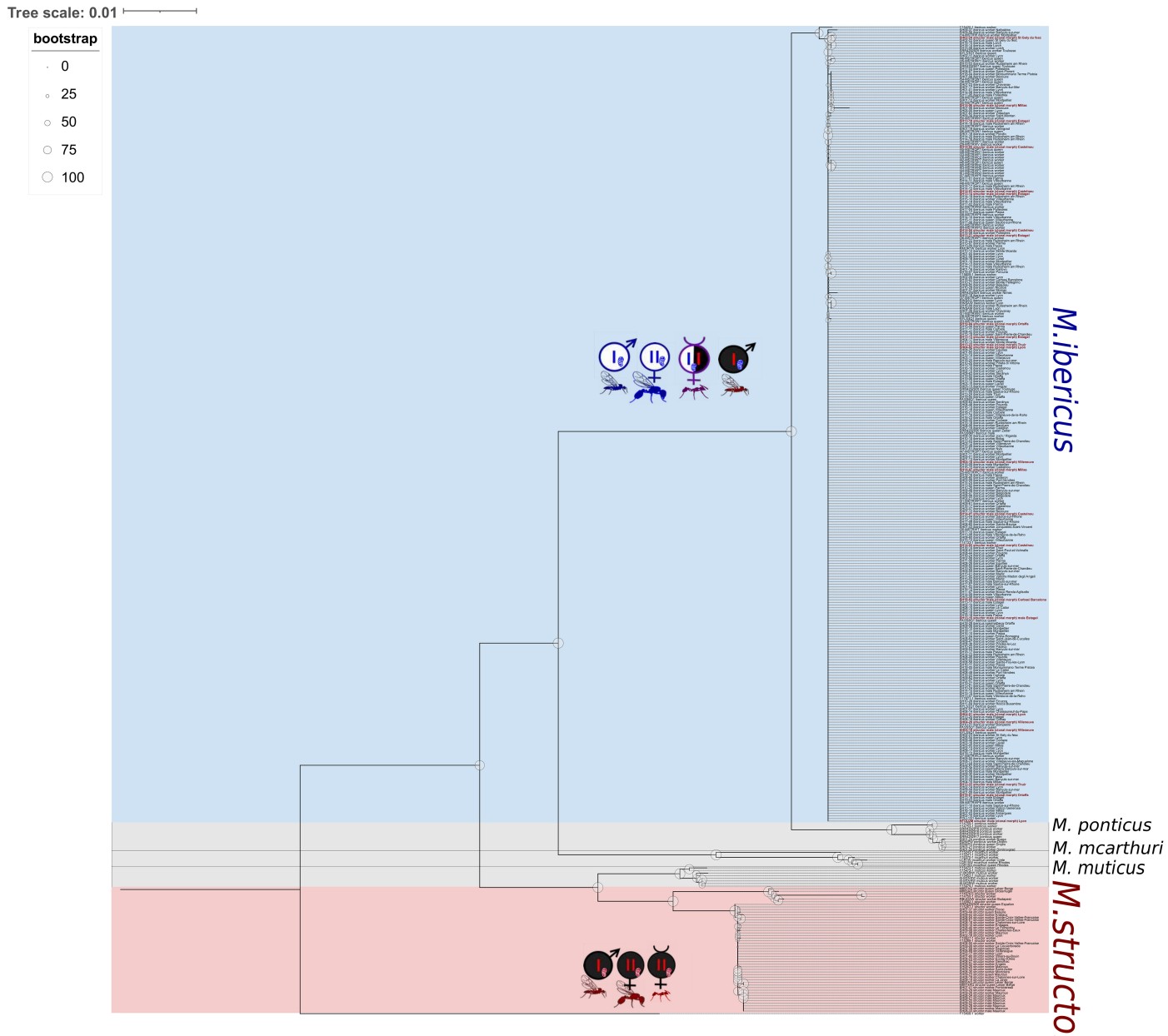

**Extended Data Fig. 2 | Mitochondrial phylogeny of 380 *Messor* individuals.**
The topology of the tree was produced from a supermatrix of 2,585 sites using *IQ-TREE* (TIM2 + F + I after model selection, 1,000 ultrafast bootstraps). *M. structor* (clonal morph) are highlighted in red in the tree. *M. ibericus* and *M. structor* clades are illustrated with their respective castes. Red and blue bars/mitochondria indicate respectively *M. structor* and *M. ibericus* nuclear/mitochondrial genomes.

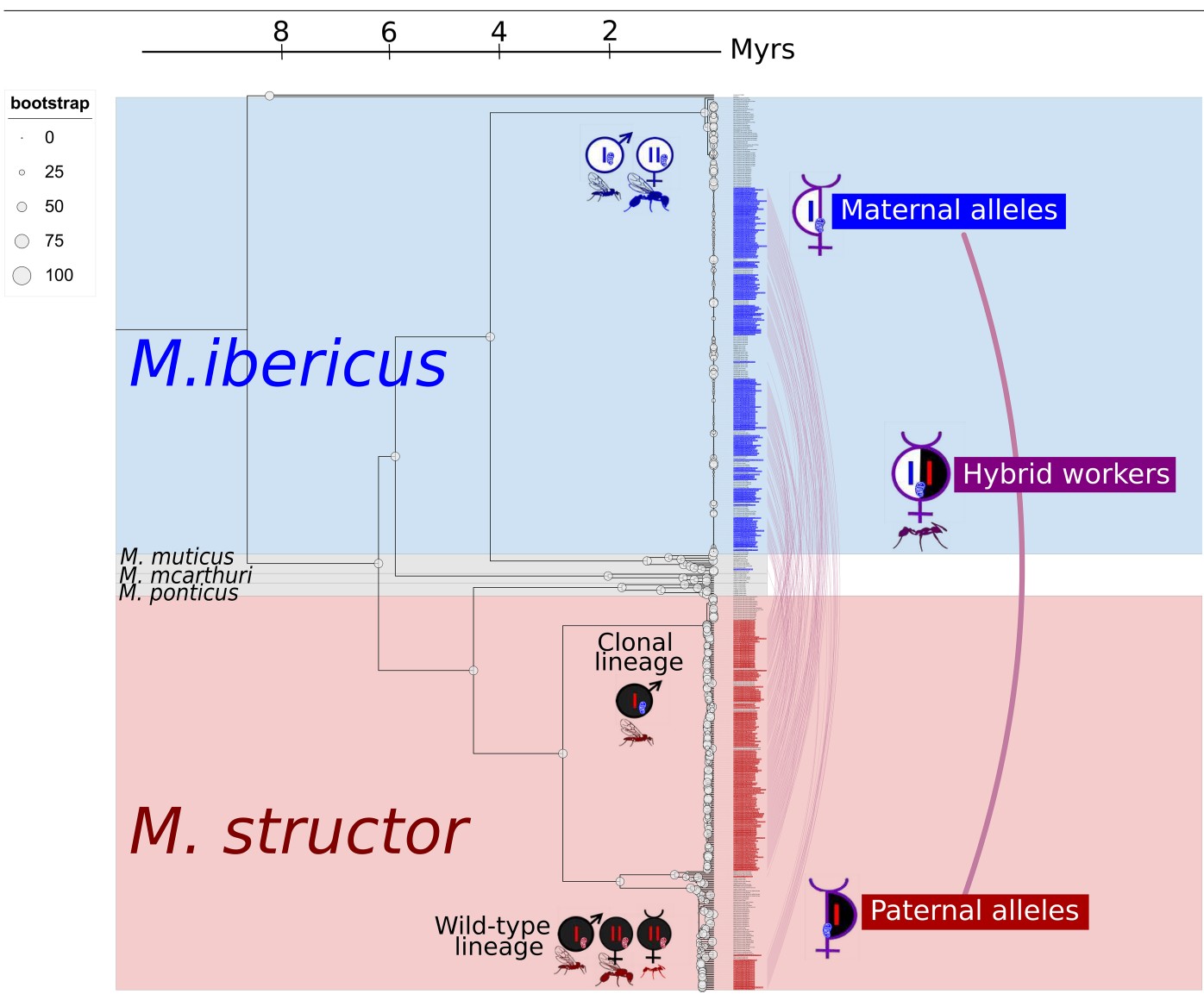

**Extended Data Fig. 3 | Molecular phylogeny of hybrid and non-hybrid individuals.** The topology of the tree was produced from a supermatrix of 1,089,038 sites using *IQ-TREE* (model GTR + I + F + G4, 1,000 ultrafast bootstraps). Hybrid genomes have been separated in different sequences (*i.e.* maternal and paternal alleles) and are highlighted in blue and red. Maternal and paternal alleles that belong to the same hybrid individual are linked by a purple curve. *M. ibericus* and *M. structor* clades are illustrated with their respective castes. Red and blue bars/mitochondria indicate respectively *M. structor* and *M. ibericus* nuclear/mitochondrial genomes. Two distant outgroups of the tree (*Aphaenogaster floridana* and *Acromyrmex echinatior*) were removed from the figure for better readability.

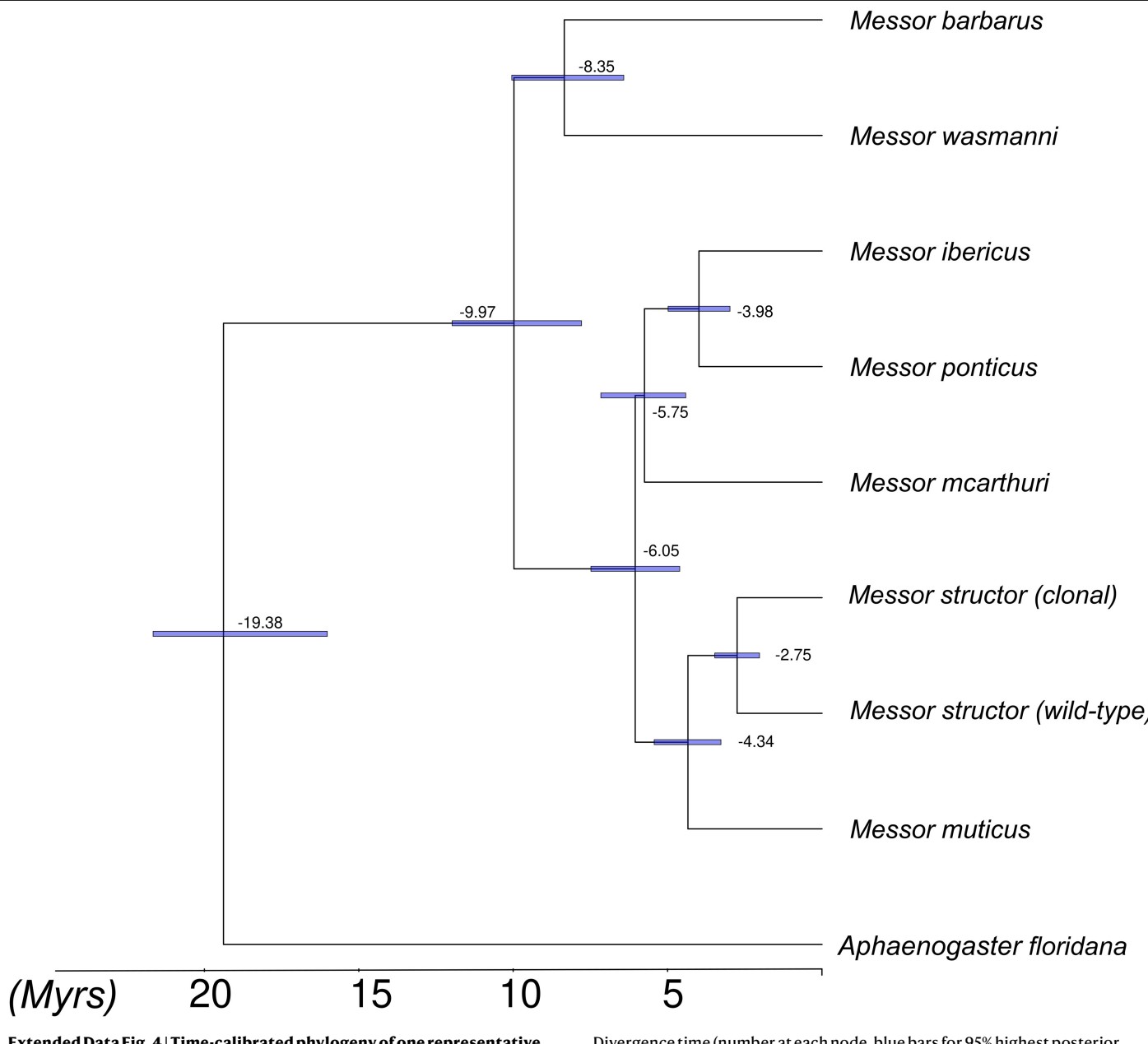

**Extended Data Fig. 4 | Time-calibrated phylogeny of one representative individual per *Messor* species.** The topology of the tree was produced from a supermatrix of 6,089,069 sites using *IQ-TREE* (model GTR + I + F + G4, 1,000 ultrafast bootstraps). All nodes have maximal bootstrap support (100). Divergence time (number at each node, blue bars for 95% highest posterior density) were estimated with *MCMCtree* and have been used to calibrate Extended Data Figs. 1–3.

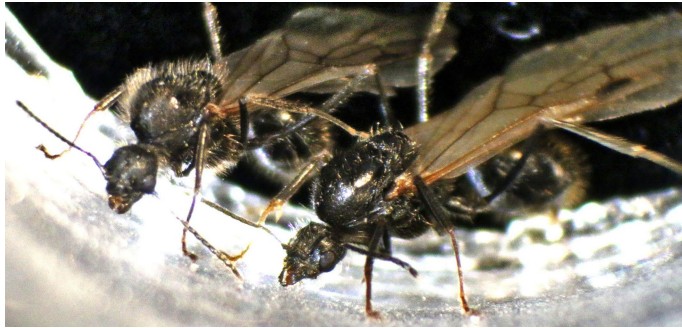

**Extended Data Fig. 5 | Picture of lived *M. ibericus* and *M. structor* males laid in the same colony.** The colony was maintained in artificial conditions for 19 months (ID MOMA1) after queen sampling in the Bois de Montmaur, in Montpellier, France. The colony was maintained at 28 °C, fed with grass seeds. Broods have been checked weekly, one adult male of each species emerged after 19 months. *M. ibericus* is here on the left, *M. structor* on the right. Species are easy to recognize with the naked eye, with clear visible differences in terms of pilosity and shape of the thorax.

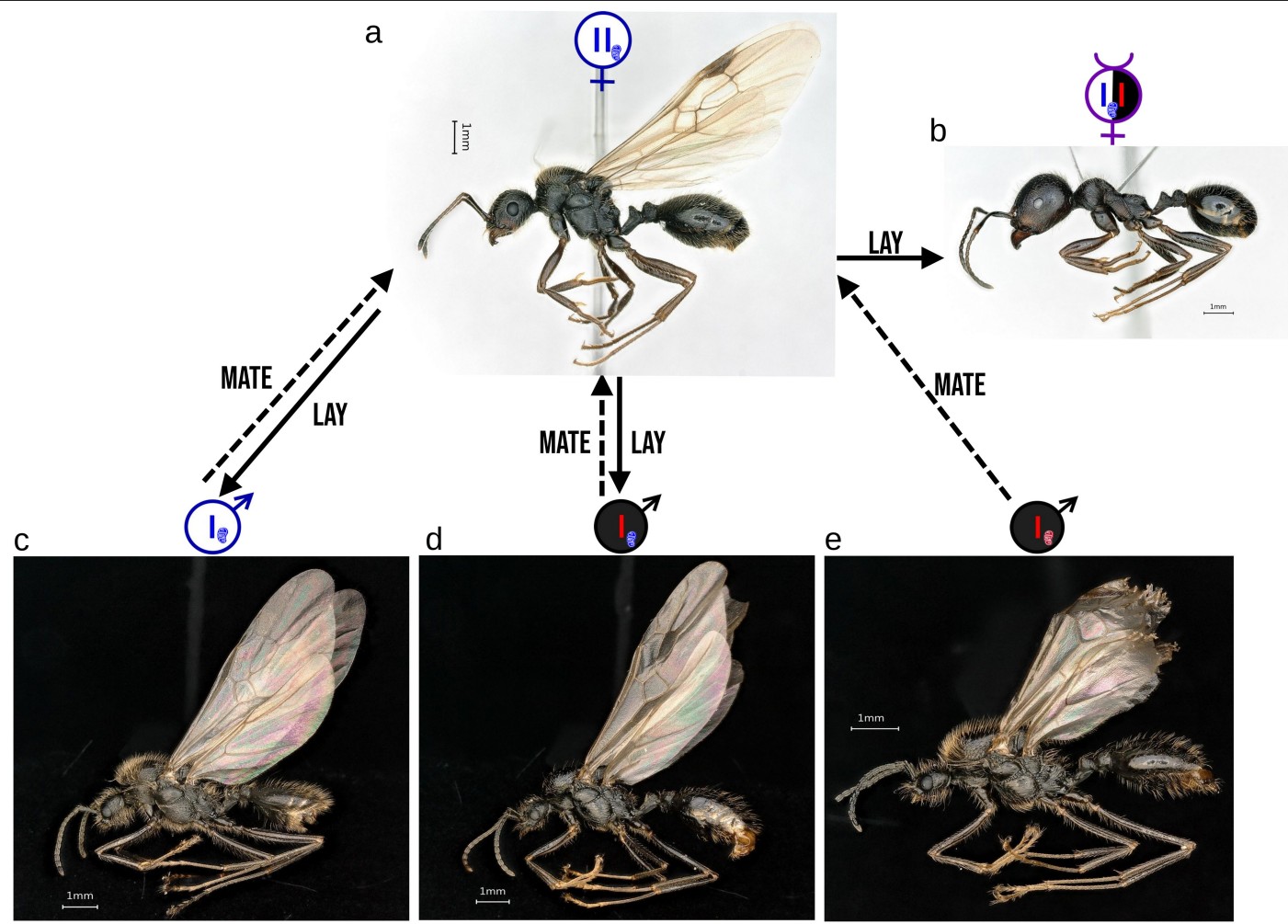

**Extended Data Fig. 6 | Pictures of all caste/sex/species involved in *M. ibericus* colonies. (a)** *M. ibericus* queen. **(b)** *M. ibericus* workers (*ibericus x structor* hybrids). **(c)** *M. ibericus* male. **(d)** *M. structor* (clonal) male. **(e)** *M. structor* (wild-type) male.

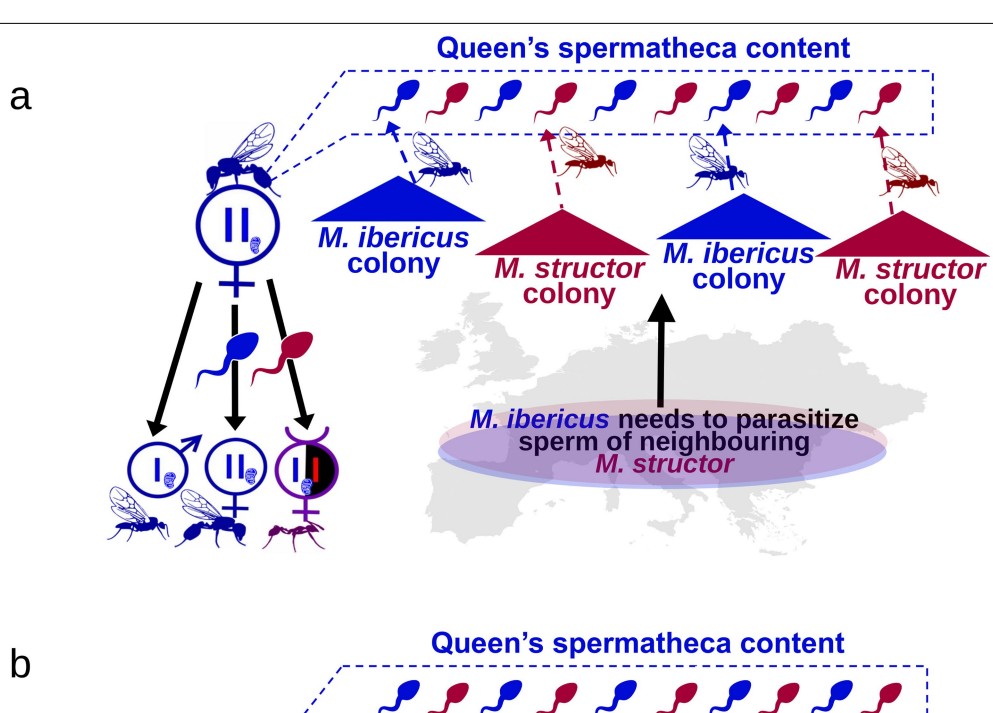

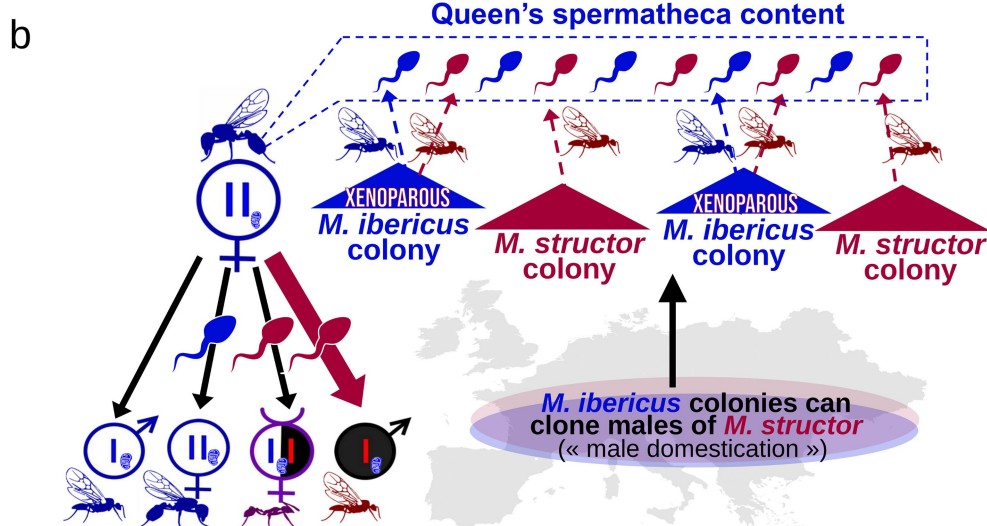

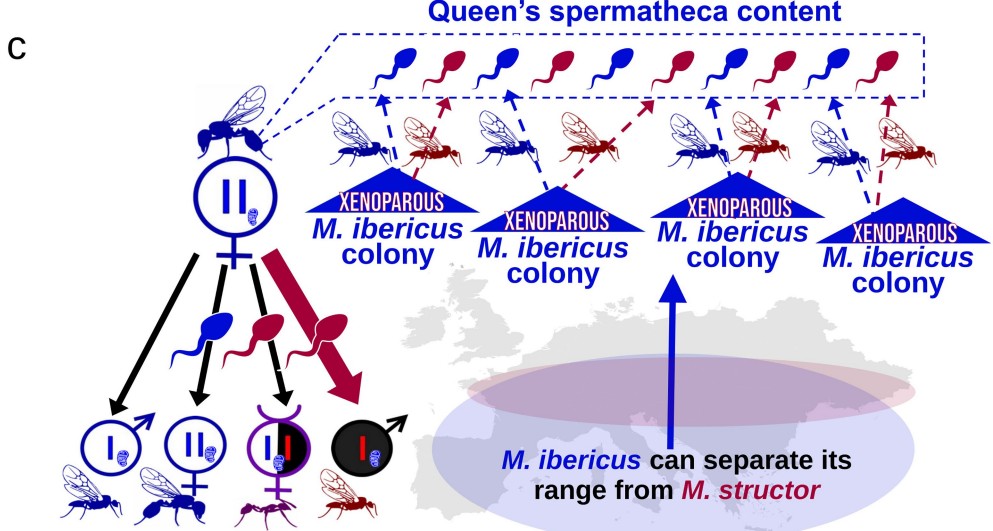

**Extended Data Fig. 7** | See next page for caption.

**Extended Data Fig. 7 | Schematic representation of geographical constraint release following male domestication. a**, *M. ibericus* queens require parasitizing sperm of *M. structor* to produce workers, *i.e.* obligate sperm parasitism. This is the initial situation, which is typically found in several other harvester ants[12,17,18,32]. **b**, *M. ibericus* queens can lay *M. structor* males cloned from the spermatozoids they regularly store in their spermatheca. This is akin to domestication, as *M. ibericus* favors the reproduction of a species they first exploited from the wild[34]. This intermediary situation is supported by our results, where areas in which both species still co-occur display *M. ibericus* workers fathered by either "domesticated" males (*i.e. M. structor* produced in "xenoparous" colonies of *M. ibericus*) or wild males (*i.e. M. structor* males produced in their own species' colonies) (Fig. 3). **c**, *M. ibericus* can sustain a domesticated clonal lineage independently by producing *M. structor* males, which serve as a sperm source. This enables the production of new workers and males in subsequent generations, allowing *M. ibericus* to invade areas where *M. structor* does not naturally occur. This is a strict "xenoparous" reproductive mode, meaning that all workers are fathered by "domesticated" males laid by *M. ibericus* queens. This situation is widespread across Mediterranean Europe (Fig. 3, Supplementary Table 1 and Extended Data Fig. 3).

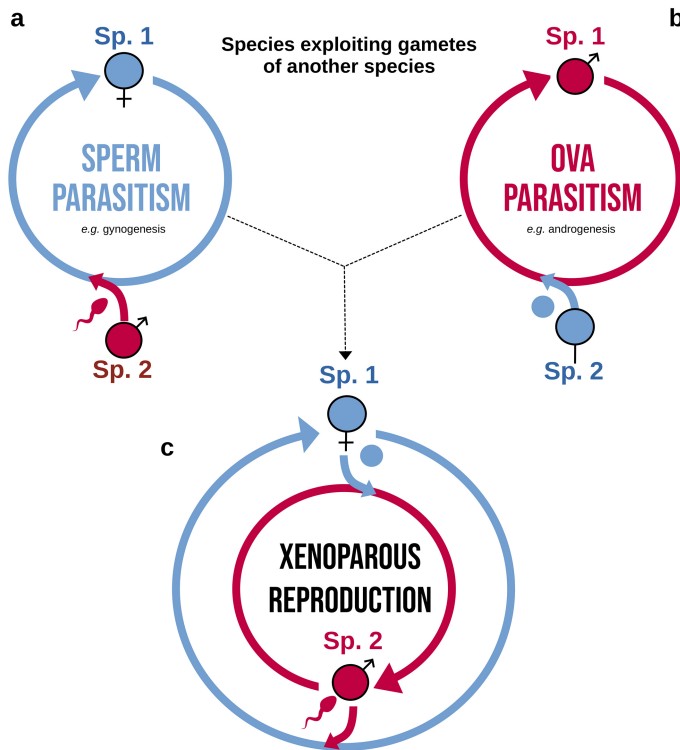

**a** Sp. 1 — Species exploiting gametes of another species — **b** Sp. 1

SPERM PARASITISM
*e.g.* gynogenesis
Sp. 2

OVA PARASITISM
*e.g.* androgenesis
Sp. 1 / Sp. 2

**c** Sp. 1

XENOPAROUS REPRODUCTION
Sp. 2

Species sexually binded into a
self-sufficient unit of selection

**Extended Data Fig. 8 | From sexual parasitisms to xenoparous reproduction.**
Species that act as sexual parasites use another species's gametes to propagate
their own genome[3]. Such sexual parasitism can be divided into two main
categories illustrated at the top of this figure (a and b). Xenoparous reproduction
(this study) is illustrated at the bottom of the figure (c). **a, Sperm parasitism**.
A female exploits the sperm of another species to reproduce. The most typical
form of sperm parasitism is gynogenesis, where females reproduce asexually
but require contact with sperm to initiate embryogenesis. After contact, the
host male's genome is immediately excluded[88]. In contrast, this exclusion is
delayed to the next generation in hybridogenesis. In this alternate form of sperm
parasitism, the male genome is present in the somatic cells of the parasite
offspring but excluded during meiosis when germinal cells are produced[89].
In social insects, hybridogenesis refers to cases where the male genome is
present in workers (*i.e.* somatic cells of a "superorganism") but excluded from
reproductive individuals (*i.e.* germinal cells of a "superorganism"). Such social
hybridogenesis is common in the *Messor* genus[12] and is the sperm parasitism at
the origin of xenoparous reproduction in *Messor ibericus* (see Fig. 3a). **b, Ova
parasitism**. A male exploits the ova of another species to reproduce. Typically
referred as male cloning, androgenesis is the flip side of gynogenesis, where
offspring only carry the nuclear genome of their father. It is excessively rare
compared with gynogenesis and typically occurs in a hermaphrodite context
with exceptional inter-specific parasitism (*Cupressus dupreziana* conifers or
*Corbicula* clams)[25–27]. To date, male clonality is restricted to intraspecific cases
in ants[29–31]. **c, Xenoparous reproduction**. Here, females need to produce males
of the other species, as both species rely on each other's gametes. This combines
elements of both sexual parasitism forms, but with a key difference: females are
not victims of ova parasitism but require it, as it is the case for males regarding
sperm parasitism. In the sole case known so far (this study) **sp1 females** "use"
**sp2 male** sperm for worker production, while **sp2 males** "use" **sp1 female** ova
for cloning. Since **sp1 females** and **sp2 males** share the same colonies, both
species benefit from having their gametes exploited by the other, as both
require workers and males. By integrating sperm and ova parasitism into a
single superorganism, such a "sexual domestication" neutralizes the sex war, as
both species are trapped into a self-sufficient unit of selection. As it results in a
cohesive reproductive unit of two species, evolution toward xenoparity can be
qualified as an evolutionary transition in individuality[49,50].

# Reporting Summary

## Statistics

For all statistical analyses, confirm that the following items are present in the figure legend, table legend, main text, or Methods section.

| n/a | Confirmed | |
|---|---|---|
| ☐ | ☒ | The exact sample size (*n*) for each experimental group/condition, given as a discrete number and unit of measurement |
| ☒ | ☐ | A statement on whether measurements were taken from distinct samples or whether the same sample was measured repeatedly |
| ☐ | ☒ | The statistical test(s) used AND whether they are one- or two-sided *Only common tests should be described solely by name; describe more complex techniques in the Methods section.* |
| ☒ | ☐ | A description of all covariates tested |
| ☒ | ☐ | A description of any assumptions or corrections, such as tests of normality and adjustment for multiple comparisons |
| ☐ | ☒ | A full description of the statistical parameters including central tendency (e.g. means) or other basic estimates (e.g. regression coefficient) AND variation (e.g. standard deviation) or associated estimates of uncertainty (e.g. confidence intervals) |
| ☐ | ☒ | For null hypothesis testing, the test statistic (e.g. *F*, *t*, *r*) with confidence intervals, effect sizes, degrees of freedom and *P* value noted *Give P values as exact values whenever suitable.* |
| ☐ | ☒ | For Bayesian analysis, information on the choice of priors and Markov chain Monte Carlo settings |
| ☒ | ☐ | For hierarchical and complex designs, identification of the appropriate level for tests and full reporting of outcomes |
| ☒ | ☐ | Estimates of effect sizes (e.g. Cohen's *d*, Pearson's *r*), indicating how they were calculated |

*Our web collection on statistics for biologists contains articles on many of the points above.*

## Software and code

Policy information about availability of computer code

| Data collection | No software was used for data collection |
|---|---|
| Data analysis | Data were analysed on R v. 4.1.2. All other softwares used for analysing genetic data are described in the Material and Methods of the article, including: Wtbg2 v. 2.5, MaSurCA v. 3.4.1, NextPolish v. 1.3.1, RagTag v. 1.0.2, TGS-GapCloser v. 1.1.1, QUAST v. 5.0, BUSCO v. 4.0.5, fastp v. 0.23.2, BWA-MEM2 v. 2.2.1, GATK v. 4.3, vcftool v. 0.1.16, MAFFT v. 7.490, trimal v. 1.4, IQ-TREE v. 2.07/v. 2.12/v. 2.2.2.7, PAML v. 4.10.7, Tracer v. 1.7.2, PLINK v. 1.90b6.21, fastStructure v. 1.0 (includes chooseK.py), dNdSpiNpiS v. 1.0, tr2 v. 1, SODA v. 1.0.2, aphid v. 0.11. Scripts used for producing the results of the study are available in the Zenodo repository at the following URL: https://zenodo.org/records/11506545 |

For manuscripts utilizing custom algorithms or software that are central to the research but not yet described in published literature, software must be made available to editors and reviewers. We strongly encourage code deposition in a community repository (e.g. GitHub). See the Nature Portfolio guidelines for submitting code & software for further information.

## Data

Policy information about availability of data

All manuscripts must include a data availability statement. This statement should provide the following information, where applicable:
- Accession codes, unique identifiers, or web links for publicly available datasets
- A description of any restrictions on data availability
- For clinical datasets or third party data, please ensure that the statement adheres to our policy

Raw reads of genetic data are deposited on NCBI under the Project ID PRJNA1145159 (available with publication). SRA IDs and all data supporting the results of the study are indicated for each sample in Table S1. Reference genomes, genetic variation data, phylogenetic analyses and scripts used for producing the results of the study are available in the zenodo repository at the following URL: https://zenodo.org/records/11506545.  We also used the following dataset from the orthoDB database (https://www.orthodb.org/): hymenoptera_odb10 (https://busco-data.ezlab.org/v5/data/lineages/hymenoptera_odb10.2024-01-08.tar.gz) and metazoa_odb10 (https://busco-data.ezlab.org/v5/data/lineages/metazoa_odb10.2024-01-08.tar.gz).

## Research involving human participants, their data, or biological material

Policy information about studies with human participants or human data. See also policy information about sex, gender (identity/presentation), and sexual orientation and race, ethnicity and racism.

| | |
|---|---|
| Reporting on sex and gender | N/A |
| Reporting on race, ethnicity, or other socially relevant groupings | N/A |
| Population characteristics | N/A |
| Recruitment | N/A |
| Ethics oversight | N/A |

Note that full information on the approval of the study protocol must also be provided in the manuscript.

# Field-specific reporting

Please select the one below that is the best fit for your research. If you are not sure, read the appropriate sections before making your selection.

☐ Life sciences ☐ Behavioural & social sciences ☒ Ecological, evolutionary & environmental sciences

For a reference copy of the document with all sections, see nature.com/documents/nr-reporting-summary-flat.pdf

# Ecological, evolutionary & environmental sciences study design

All studies must disclose on these points even when the disclosure is negative.

| | |
|---|---|
| Study description | DNA was extracted from ant individuals. Shotgun sequencing data were generated and aligned to a reference genome of Messor ibericus to analyse it. |
| Research sample | 390 ant individuals were analysed in this study, from which 326 whole genomes and 52 transcriptomes have been newly sequenced. These samples range across Europe (from Spain to Turkey). Details are listed in Table S1. |
| Sampling strategy | Genome-wide data for 390 individuals is considered as above average for population genetics. We focused particularly the sampling on the two species of interest (289 Messor ibericus and 77 Messor structor). |
| Data collection | Genomic data was collected according to laboratory protocols to minimise contamination (details in Methods Section). DNA and RNA libraries were sequenced via Illumina sequencing. |
| Timing and spatial scale | Samples range across Europe and were collected between 2000 and 2022. |
| Data exclusions | No data were excluded. |
| Reproducibility | Genome and transcriptome sequencing has been repeated several times and all necessary informations are available in the Methods Section. Egg genotyping has also been repeated several time successfully and the Methods Section provide all necessary information. |
| Randomization | Randomization was not applicable in this study. Samples were grouped into their species origin identified morphologically then genetically. |

| Blinding | DNA/RNA extraction and library preparations have been performed without initially knowing the species of origin. |
|---|---|

Did the study involve field work?  ☒ Yes  ☐ No

## Field work, collection and transport

| Field conditions | Performed in spring, summer and autumn across Europe, weather usually sunny, often after a rainfall to maximize the probability to find reproductive individuals. |
|---|---|
| Location | All locations are described in Table S1. |
| Access & import/export | All field samples were collected on public property or on private property with permission of the land owner. None of the species collected were endangered or collected on protected lands. |
| Disturbance | Not applicable. |

# Reporting for specific materials, systems and methods

We require information from authors about some types of materials, experimental systems and methods used in many studies. Here, indicate whether each material, system or method listed is relevant to your study. If you are not sure if a list item applies to your research, read the appropriate section before selecting a response.

## Materials & experimental systems

| n/a | Involved in the study |
|---|---|
| ☒ | ☐ Antibodies |
| ☒ | ☐ Eukaryotic cell lines |
| ☒ | ☐ Palaeontology and archaeology |
| ☐ | ☒ Animals and other organisms |
| ☒ | ☐ Clinical data |
| ☒ | ☐ Dual use research of concern |
| ☒ | ☐ Plants |

## Methods

| n/a | Involved in the study |
|---|---|
| ☒ | ☐ ChIP-seq |
| ☒ | ☐ Flow cytometry |
| ☒ | ☐ MRI-based neuroimaging |

## Animals and other research organisms

Policy information about studies involving animals; ARRIVE guidelines recommended for reporting animal research, and Sex and Gender in Research

| Laboratory animals | This study did not involve laboratory animals. |
|---|---|
| Wild animals | Individuals of various species (M. ibericus, M. structor, M. ponticus, M. mcarthuri and M. muticus), caste (queens, males, workers) and ages (<1 year for males and isolated queens, >1 year for mature colonies) were captured in their natural habitat with tweezers and transported in plastic boxes. Some colonies with queens were maintained in artificial nests in the lab and were monitored for up to 36 months. Other individuals were stored in eppendorf filled with EtOH for conserving their DNA before sequencing their genome. No individuals were released. |
| Reporting on sex | Sex was identified morphologically and is reported in Supplementary Table 1. |
| Field-collected samples | Collected ants were reared in fluoned boxes (queens, workers, males) in a room at 28°C with 40% humidity (no light/photoperiod to mimick their underground lifestyle). Nests were cleaned twice a week and they were provided with water and grass seeds. Colonies are either still maintained in the lab or have been maintained until their natural death.<br><br>None of the species collected are endangered as determined by their absence on the IUCN Red List of Threatened Species. |
| Ethics oversight | No ethical approval is required for research on non-endangered invertebrates. |

Note that full information on the approval of the study protocol must also be provided in the manuscript.

## Plants

Seed stocks

N/A

Novel plant genotypes

N//A

Authentication

N/A

