## [Peer Review File · Nature]

One mother for two species: obligate cross-species cloning in ants

Corresponding Author: Dr Jonathan Romiguier

Version 1:

Reviewer comments:

Referee #1

(Remarks to the Author)

The manuscript by Juvé et al. report the discovery of an ant species (*Messor ibericus*) where the queen of the colony lays males from two distinct species (*Messor ibericus* and *Messor structor*). In support of this finding, the Authors use a population genetics and taxonomic approach to show that males from the same mother exhibit distinct genomes and morphologies, belonging to species that diverged over 5 million years ago. The Authors argue that this life cycle has evolved because individuals in the worker caste can only be produced as hybrids between these two species, and therefore, in geographic regions where these two species do not overlap, queens must “clone” males of another species so that they can use the sperm of these males to produce the obligately hybrid worker caste. The authors argue that the evolutionary history of this system first appeared as sexual parasitism and subsequently evolved into a natural case of “cross-species cloning.”

This is an arresting observation that challenges basic assumptions about reproduction within a species. Furthermore, although I am not an expert in population genomics, from my reading, I believe the data support the general conclusions being made in the manuscript. Based on these two conclusions, I think this paper is a worthy candidate for publication in Nature.

Here are some more detailed comments that I hope the Authors can address to improve the interpretations of the manuscript:

Concern # 1:

First, while I believe that the patterns of transmission of this life-cycle is convincing, the mechanistic processes underpinning the “cloning of males” during the life cycle remains unknown. After reading the manuscript several times, it still remains unclear to me how *M. ibericus* queens are acquiring the paternal gene to lay *M. structor* males (which are then used to produce hybrid workers). *M. ibericus* queens are presumably only mating with *M. ibericus* males in areas where the two species (*M. ibericus* and *M. structor*) do not overlap. Therefore the male sperm is 100% *M. ibericus*. How then, mechanistically speaking, are they actually cloning the *M. structor* sperm of males? In other words, from where are they getting the paternal gene copy of *M. structor* if there is no sperm or if there is sperm, where is it coming from? To this reviewer, the answers to these questions were not clear from the manuscript.

Furthermore, the Authors use the term “cloning” but am not so sure if this means the same thing as in another clonal (ant) species. Given that there is already a mating system with fertilization using sperm of males from *M. ibericus*, does cloning work in the same way as in other (ant) species that clone? Are the authors assuming this process is conserved (the same)? However, given the complexities of this system, I am not so sure that the Authors can assume this.

Therefore, the Authors should remain neutral about what is happening at the mechanistic level and present alternative hypotheses from the literature about how this could potentially be happening. Also, the Authors should be prepared, depending on what they find, to abandon the “cloning” “cross-species cloning” or “males hijack eggs to clone ...”. These statements are laden with mechanistic assumptions that I believe the Authors have no evidence for.

Concern # 2:

The methods are complex and the writing in the manuscript is dense. In general, I had to work hard to understand which analysis was testing what. So, I think it will be easier if the Authors, step by step, associated the method used for each result and tell the Reader what they should see, as well as what the alternative pattern would mean. It is also important for the Authors to mention the level of statistical confidence they have in each findings, which this Reviewer could not easily ascertain from the data.

Concern # 3:

The Authors never really tell us why they selected the Messor species that they did. Given their previous study on this, why did they not use any of the species from the previous study? What would they have expected to find in these other species?

Concern # 4:

References 8 & 9 are not sufficient for this statement. Please find alternative or additional references for this statement.

Concern # 5:

It would be important to spell out the logic of why the authors think they are "1st generation hybrids"

Concern # 6:

They go back and forth between individual-level and colony-level (superorganism) perspectives and interpretations. This is confusing. I think it is better to state these in one place and talk about both alternatives.

Referee #2

(Remarks to the Author)

This manuscript is clear and well-written and uses an impressive array of methods to describe a fascinating example of "sexual parasitism". Overall, I think this is very impressive work, and I wanted to be clear about that before moving on to present some critical comments. The central claim is the first known case of a "xenoparous" reproductive mode, where individuals "give birth to other species as part of their lifecycle." This novelty serves as the focus of the manuscript, so I will address it first.

While it is indeed an exciting and novel finding, I do not find it as groundbreaking as the authors suggest. For one, is egg-laying truly equivalent to giving birth? While this could be seen as a matter of semantics, I believe there is a significant distinction in the whole range of reproductive systems. Organisms exhibit a continuum of reproductive strategies, such as live birth, egg-laying, or broadcast spawning. Would the discovery of a broadcast spawner's egg developing into a different species still be considered groundbreaking? Or what if gametes from two different broadcast spawning species fused, resulting in an offspring of only one of the species? What about a case where a zygote developed into a different species than the father who fertilised the egg? This phenomenon is already known (gynogenesis), and seems analogous, the main difference being that in one system the maternal gamete is parasitized, while in the other the paternal gamete is paratitized. And as the authors themselves mention in the discussion, cross-species cloning via maternal gametes is already known too, although not in as complex a system as described here.

The point I am trying to convey is that there exists a wide range of unusual "sexually parasitic" reproductive modes, and while the system described in this study is striking, it does not seem entirely distinct from what is already known. My understanding is that the crucial element here is that a fertilized gamete from a mother can produce progeny of a different species, regardless of the exact method of offspring delivery, and that this different species is necessary for maintenance of the first species. This concept closely mirrors cases like gynogenesis, where a gamete from a father fuses with an egg to produce progeny of a different species, independent of the delivery process.

While I do not think the central thesis is wrong, I do think it should be more fairly integrated with current understanding of such parasitic reproductive systems.

The discussion is quite speculative and, in some respects, exaggerates the meaning or novelty of the results. Here the authors briefly discuss androgenesis, but quite superficially and it is not clear what the implication is. As I understand it, the message is that cross-species cloning via females is known in other species too, but it is not obligate and is thus less striking than the system described here. This is of course true to an extent, but nevertheless, it does decrease the novelty of the current results in a similar way to gynogenesis discussed above.

The authors then describe xenoparity as a symbiotic mutualism from an evolutionary perspective, and the males in the system as a perfected form of male parasites. Clonal males are discussed as organelles at the superorganism level, and the manuscript closes by stating that this xenoparous colonial lifeform challenges usual boundaries of organismic complexity. All of this is interesting, but it is not really backed up by the results. That the system has evolved as a symbiotic mutualism seems quite uncertain to me, or that the males can really be considered perfected parasites. If I was to speculate, I would guess the most likely explanation for such a system is a type of evolutionary trap where 'the best of a bad job' (once a dependence on males of a different species has arisen perhaps partly by chance events) is maintained. It is not clear to me how such a system might be beneficial to the host species, over the 'typical' way of making workers and males. Are the males perfected parasites? It seems more like they have been co-opted into the life-cycle of the host. Do they have the possibility of their spreading their genes outside of this domesticated situation? If not, is it again more like an evolutionary trap than perfected parasitism? Finally, I don't think the system really challenges usual boundaries of organismic complexity. It does have a novel feature, but there are many other species with complex superorganism colonies that surpass the complexity of this one in some other way.

Referee #3

(Remarks to the Author)

Summary of Key Results; Originality and Significance: In their manuscript, Juvé et al. lay out a compelling case for their novel discovery of an ant that produces both “normal” offspring and heterospecific males. They show that all *Messor ibericus* workers are F1 hybrids (with *M. structor* fathers) and, in allopatric regions where heterospecific males are not otherwise present, *M. ibericus* queens produce a clonal lineage of *M. structor*-derived males. This stunning finding is laid out in a clear and convincing manner in the article. I am convinced that the main result is valid, but I suggest that the authors take one additional step to demonstrate that the clonal male lineage produced by *M. ibericus* queens is still functionally *M. structor*, as the authors claim. In addition to this major suggestion, I have some minor comments and ideas.

Data and methodology: The methods appear to be appropriate, and are clearly explained and documented.

Conclusions: The conclusions are generally robust and well supported.

Suggested improvements:

Major suggestion:

This article is framed as an instance of what the authors term “xenoparity”, wherein an individual of one species produces offspring of another species. The authors systematically demonstrate that single *M. ibericus* queens can produce two types of males: those with *M. ibericus* mitochondrial and nuclear genomes and those with *M. ibericus* mitochondrial genomes and *M. structor*-derived nuclear genomes. The latter clonal males form a sister clade to the so-called “wild type” *M. structor* males (and queens/workers and are morphologically distinct from both *M. ibericus* males and *M. structor* wild type males). Since these males father *M. ibericus* (hybrid) workers, they clearly retain viability. However, in order to demonstrate that these males are indeed *M. structor*, as the authors argue, it is necessary to demonstrate that they can still mate with *M. structor* queens and produce viable offspring (at least in the lab, if not in the field). To me, this seems like a final piece of the puzzle that is needed in order to show that these males are truly still *M. structor* and not a distinct lineage of hybrid origin in their own right. To be clear, even without this addition, I still think that the results of this study are very exciting. However, if the authors do not demonstrate that the *M. ibericus*-produced heterospecific males can still successfully mate with *M. structor* queens, I would suggest softening the language (perhaps by referring to these males as having *M. structor*-derived nuclear genomes, as I do above, or similar).

Minor comments and suggestions:

Line 26: individuals -> offspring?

Line 39: Such a complexity -> Such complexity; same line, consider removing “need to additionally”

Line 42: contrasts -> variation? Or distinctiveness?; same line, consider omitting “required to be”

Line 46: omit “yet”?

Line 49: consider removing “exclusively”

Line 51: I realize that this “symbiosis” idea was explained later, but from the context provided so far, it is not apparent that *M. structor* requires the gamete of *M. ibericus* (the *M. ibericus*-produced heterospecific males require *ibericus* ova, but the *M. structor* species does not require a heterospecific gamete to persist”. This sentence really confused me on the first read through.

Line 58: consider rewording or at least moving or removing “females”- this formulation was confusing.

Lines 58-61: I noticed that the total sample size presented here did not match the 390 individuals mentioned above.

Line 78: other -> another

Line 80: comes as -> is

Line 123: of -> for

Line 145: omit the first instance of “two”?

Lines 185-188: This is an interesting idea, but I’m not sure that the authors can conclude that the morphological difference resulted from the loss of genetic diversity. Couldn’t there also be an interaction between genetic markers in the nuclear and mitochondrial genomes, plasticity due to different rearing conditions, or any number of other causes of this difference? The authors mention some alternative possibilities later in the paragraph, but I found this sentence to be a little misleading.

Line 189: “compared to the others” – which others do the authors mean?

Line 212: awkward wording (In particular, “the need to clone” seems a little strange)

Line 221: appears as -> appears to be a

Lines 231-244: Very cool parallels!

Lines 401-405: This phasing approach seems a little simplistic and likely to lead to incorrect inference, as described here: “Based on sampling and previous results (mitochondrial phylogeny, Fig. S1), we know that *M. ibericus* is the maternal species of hybrid workers, so that the alleles that do not match a *M. ibericus* genome belong to the paternal species. We exploited this by writing a python script comparing each SNP of each hybrid worker with variants of a reference maternal genome (*M. ibericus* queen genome with the highest coverage, SH19-06).” Overall, this approach probably gets you close enough, but the statement that I italicized seems problematic to me if there is substantial genetic variation across the sampled range of *M. ibericus*.

References: To the extent that I am familiar with this subfield, it appears that previous work is cited appropriately.

Clarity and context: I found the article to be written in a clear and compelling way.

Referee #4

(Remarks to the Author)

Review of manuscript "One mother for two species: obligate cross-species cloning in ants" by Juve et al.

The research team reports an extraordinary type of reproduction in ants. Apparently, the queen produces males of two different species. And then has to mate with males of these two species in order to produce a functional colony because 'hybrid' workers are necessary to success.

I found this work very interesting. Several ant species have been found to engage in unusual reproductive mechanisms in the past 20 years. But this new proposed mechanism is quite amazing and unique.

However, the one question I had was about mechanism. What exactly is happening from a developmental and reproductive standpoint to allow this to occur. Queens are apparently not 'hybrid' so are queens polyandrous? And then the male is essentially cloning itself through its sperm as the authors suggest? I wasn't sure exactly how this system could occur. But I did feel that some greater insight into mechanism would be important to having a complete story about this extraordinary system.

Version 2:

Reviewer comments:

Referee #1

(Remarks to the Author)

The revised version of "One mother for two species: obligate cross-species cloning in ants" by Juve et al. has much improved. The Authors have addressed all my concerns from the previous round of review. I have just have three points to help improve the manuscript before publication:

Point # 1:

The reader would benefit from more background information about the selective advantage of maintaining hybrid workers for the colony. Without this key piece of information, it is hard to understand why they would be maintained in the colony. For example, on Lines 168-169, the Authors state there are many well-described examples of queens using other species' sperm to produce their workers (and suggest this is the ancestral state of their study system). They cite many references describing hybrid workers, but to someone not familiar with this literature it is a very surprising finding in itself, and it would be very helpful to briefly mention why such a system might evolve (why use another species' sperm vs your own?).

Point # 2:

Once again, there is tension in the manuscript between levels of selection (individual versus colony). To me, the most exciting interpretation is that the evolution of xenoparity is a novel kind of major evolutionary transition in individuality (METI). This implication by itself, in my opinion, is enough to make it Nature worthy. Yet, the Authors give it only 1 sentence in the discussion. Furthermore, this interpretation of the data appears to be at odds with individual level explanations and associated use of terminology. For example, they frame the ancestral state as the queen "exploiting" another species' sperm to make workers (implying using heterospecific sperm has benefits for worker production). But in paragraph beginning line 245 it is framed much more negatively ("locked into obligate sperm parasitism", "best of a bad situation"). Of course, there are other loaded terms used, like "hijacking" (which I don't like) and domestication, all of which are individual-level explanations that assign intention or agency to the individual workers reproductive organs, which don't really make that much sense from the perspective of developmental biology or colony-level / systems evolution. From the superorganismal perspective, it would seem more reasonable to interpret the ancestral state as a mutualism not parasitism. What I mean is, there could be a mutual advantage, where hybrid workers are more fit and the second species benefits because its DNA is getting passed on (even if by another species). So I believe the authors should be a bit more open about the language / interpretations when reconstructing the evolutionary history and describe multiple possibilities except for just one. For example, it could have been any one of these evolutionary pathways:

Sperm parasitism → cross-species cloning / METI

Sperm mutualism → cross-species cloning / METI

Sperm parasitism → mutualism → cross-species cloning / METI

Point #3:

I appreciate the Authors providing further information (or at least as much as they could provide) about the mechanisms. This will be an interesting mechanism to crack in future studies, and I feel this would be worthy of saying in a single sentence at the end of the discussion. More importantly, however, I liked the figure they provided in the rebuttal more than the one they have in the paper and think they should consider switching it in.

Referee #2

(Remarks to the Author)

I reviewed an earlier version of this manuscript, which was subsequently rejected based on the comments of multiple referees. I have nevertheless been asked to review a resubmission of the manuscript, and to prevent delays in the decision I will keep my review brief. I have read the revised manuscript and the responses and by and large, my original review stands. The manuscript is very interesting and as far as I can assess the methodology, quite convincing. Its groundbreaking nature is a matter of interpretation and perspective, and while publication in a top journal is warranted, in my opinion the case for publication in Nature is marginal.

Referee #3

(Remarks to the Author)

Overall, the authors have done an excellent job in addressing reviewer comments and suggestions, and their effort is reflected in a more compelling and clear manuscript. I remain convinced that this article is of sufficient quality and general interest to warrant publication in Nature. I have several remaining wording suggestions- most are minor or are subjective comments, so I present them below in line number order:

Line 26: should be "same-species OFFSPRING" (no s) – the word offspring can be singular or plural.

Line 28: lifecycle should be two words (life cycle)

Line 50: such a -> this

Lines 68-70: technically, I think that determining that the *M. ibericus* workers align with *M. ibericus* queens in the mitochondrial phylogeny confirms that they have mothers with *M. ibericus* mitogenomes, but doesn't rule out the possibility of some sort of clonal reproduction by workers. Without the other evidence provided later in the manuscript, this does not confirm that the *M. ibericus* queens mothered the workers. Rephrase this sentence to be more precise?

Lines 72-74: As above, this sentence "The resulting phylogenetic tree revealed that hybrid workers were all sired by *M. structor* males, as all paternal alleles (N=164) formed a well-supported clade with individuals of this species (Fig. S3)." does not rule out the possibility that workers reproduce clonally. I know that worker cloning would be unusual, but the reproductive system being reported is also strange. I suggest softening these statements until the additional lines of evidence supporting the xenoparity hypothesis are laid out.

Lines 115-119 and supp info section 4: Given the presently available information about the clonal males, I still think that their species status is unclear. Perhaps defining the species concept that the authors are employing will help. Based on the biological species concept, we can't assess the species status (there is no evidence about whether the clonal males can or cannot cross with free-living *M. structor* females). Based on the morphological species concept, the clonal lineage is different from males of either species. The authors apparently rely on the phylogenetic species concept (since their tests all focus on genetic similarity of the nuclear genome). I can live with this assessment, though I note that there are numerous empirical examples of coadaptation between mitochondrial and nuclear genes (e.g. mitonuclear incompatibilities in hybridizing swordtails, <https://www.nature.com/articles/s41586-023-06895-8>); resulting behavioral or physiological differences based on the interaction between the mitochondrial and nuclear genome could affect the cohesiveness of a species.

The additional nuclear analyses are still useful, but doubting the nuclear similarity of the two groups was not the basis of my initial comment. I appreciate the authors' additional efforts, but I still think that calling these males by the name of the free-living species from which their nuclear genome was originally captured is premature. I also don't agree that the alternative (proposed in the response to reviews) is describing the clonal males as a new and distinct species at this stage (they are not capable of propagating themselves at any scale), but instead the hijacked genetic information that is propagated exclusively in *M. ibericus* colonies could be viewed as a portion of the essential genetic complement of *M. ibericus*. But as I said, as long as the authors are clear about how they are classifying the different branches of the *M. structor* species, I think this more semantic discussion can be delayed for now. The strange reproductive systems of ants will always evade some of the tenets of our classification system.

Line 136: lived -> live?

Lines 142-144: Out of curiosity, does this happen when producing conspecific males as well? In other words, have the authors investigated whether any of the *M. ibericus* males are not closely related to their mother? Not essential for this study, but it would be interesting to know more about the mechanism.

Referee #4

(Remarks to the Author)

This is really a wild discovery. I reviewed an earlier version of this manuscript. I have no further comments. Very exciting finding, the analyses are detailed, and the manuscript is well written.

Version 3:

Reviewer comments:

Referee #1

(Remarks to the Author)

The Authors have addressed all my comments from the previous round of review. As a result, the manuscript has much improved and is ready for publication in Nature.

Response to Reviewers' Comments for the Manuscript
2024-11-23818A in Nature

One mother for two species: obligate cross-species cloning in ants

Y. Juvé, C. Lutrat, A. Ha, A. Weyna, E. Lauroua, A. C. Afonso Silva, C. Roux, E. Schifani, C. Galkowski, C. Lebas, R. Allio, I. Stoyanov, N. Galtier, B.C. Schlick-Steiner, F.M. Steiner, D. Baas, B. Kaufmann, J. Romiguier

Note: To enhance the readability of this response letter, all the reviewers' comments are typeset in blue boxes. Rephrased or added sentences in the main manuscript are typeset in red. References cited in this response are listed at the end of this letter. A tracked-change version of the main text (changes highlighted in red) is provided immediately after the clean main text in the merged PDF available for review.

Referee #1

The manuscript by Juvé et al. report the discovery of an ant species (*Messor ibericus*) where the queen of the colony lays males from two distinct species (*Messor ibericus* and *Messor structor*). In support of this finding, the Authors use a population genetics and taxonomic approach to show that males from the same mother exhibit distinct genomes and morphologies, belonging to species that diverged over 5 million years ago. The Authors argue that this life cycle has evolved because individuals in the worker caste can only be produced as hybrids between these two species, and therefore, in geographic regions where these two species do not overlap, queens must “clone” males of another species so that they can use the sperm of these males to produce the obligately hybrid worker caste. The authors argue that the evolutionary history of this system first appeared as sexual parasitism and subsequently evolved into a natural case of “cross-species cloning.”

This is an arresting observation that challenges basic assumptions about reproduction within a species. Furthermore, although I am not an expert in population genomics, from my reading, I believe the data support the general conclusions being made in the manuscript. Based on these two conclusions, I think this paper is a worthy candidate for publication in Nature.

We thank the reviewer for their general positive comment and appreciate that the significance of our results is recognized as worthy of publication in Nature.

Here are some more detailed comments that I hope the Authors can address to improve the interpretations of the manuscript:

Concern # 1:

First, while I believe that the patterns of transmission of this life-cycle is convincing, the mechanistic processes underpinning the “cloning of males” during the life cycle remains unknown. After reading the manuscript several times, it still remains unclear to me how *M. ibericus* queens are acquiring the paternal gene to lay *M. structor* males (which are then used to produce hybrid workers). *M. ibericus* queens are presumably only mating with *M. ibericus* males in areas where the two species (*M. ibericus* and *M. structor*) do not overlap. Therefore the male sperm is 100% *M. ibericus*. How then, mechanistically speaking, are they actually cloning the *M. structor* sperm of males? In other words, from where are they getting the paternal gene copy of *M. structor* if there is no sperm or if there is sperm, where is it coming from? To this reviewer, the answers to these questions were not clear from the manuscript.

We thank the referee for pointing out that this aspect was unclear in the manuscript. According to our inferences, *M. ibericus* first acquired *M. structor* sperm from areas where both species used to overlap. By cloning males from this sperm, *M. ibericus* can maintain this clonal lineage independently, which serves as a sperm source. This enables the production of new workers and males in subsequent generations, allowing *M. ibericus* to invade areas where *M. structor* does not naturally occur.

To clarify this point in the discussion, we now refer to a new supplementary figure (Fig. S17) to illustrate the process. The figure and its detailed legend are provided below for convenience. We hope it will ease the understanding of the current distribution area of both species.

Fig. S17: Schematic representation of geographical constraint release following male domestication

a, *M. ibericus* queens require parasitizing sperm of *M. structor* to produce workers, *i.e.* obligate sperm parasitism. This is the initial situation, which is typically found in several other harvester ants¹⁻⁴. **b,** *M. ibericus* queens can lay *M. structor* males cloned from the spermatozooids they regularly store in their spermatheca. This is akin to domestication, as *M. ibericus* favors the reproduction of a species they first exploited from the wild⁵. This intermediary situation is supported by our results, where areas in which both species still co-occur display *M. ibericus* workers fathered by either “domesticated” males (*i.e.* *M. structor* produced in “xenoparous” colonies of *M. ibericus*) or wild

males (*i.e.* ***M. structor*** males produced in their own species' colonies) (Fig. 3). ***c.*** ***M. ibericus*** can sustain a domesticated clonal lineage independently by producing ***M. structor*** males, which serve as a sperm source. This enables the production of new workers and males in subsequent generations, allowing ***M. ibericus*** to invade areas where ***M. structor*** does not naturally occur. This is a strict "xenoparous" reproductive mode, meaning that all workers are fathered by "domesticated" males laid by ***M. ibericus*** queens. This situation is widespread across Mediterranean Europe (Fig. 3, Table S1, Fig S3).

Furthermore, the Authors use the term "cloning" but am not so sure if this means the same thing as in another clonal (ant) species. Given that there is already a mating system with fertilization using sperm of males from *M. ibericus*, does cloning work in the same way as in other (ant) species that clone? Are the authors assuming this process is conserved (the same)? However, given the complexities of this system, I am not so sure that the Authors can assume this. Therefore, the Authors should remain neutral about what is happening at the mechanistic level and present alternative hypotheses from the literature about how this could potentially be happening. Also, the Authors should be prepared, depending on what they find, to abandon the "cloning" "cross-species cloning" or "males hijack eggs to clone ...". These statements are laden with mechanistic assumptions that I believe the Authors have no evidence for.

We appreciate the reviewer's feedback regarding the use of the term "cloning". The term might be indeed confusing, as it can refer to very different processes, either artificial or natural. In the context of a "clonal species", it typically refers to female cloning through processes like **parthenogenesis** (where an unfertilized egg develops into an embryo) or **gynogenesis** (where sperm triggers egg development without contributing genetic material)^{6,7}.

The reviewer is correct when noting that we may not refer to the same process here, as we refer to male cloning rather than female cloning. Male cloning has also a more specific term, **androgenesis**, defined as a reproductive process in which a male is the sole source of nuclear genetic material in the embryo⁸. We now clarify this point by mentioning this specific term in the main text. Following the referee's suggestion, we also briefly mention the main mechanistic hypotheses of androgenesis from the literature. Since the term "cloning" is generic and typically used to refer to androgenesis⁸, we have retained its use after providing this clarification.

Here are our modifications in the main text (line 139):

This observation points to androgenesis (*i.e.* male clonality), whereby a male provides the sole source of nuclear genetic material for the embryo⁸. Embryos devoid of maternal DNA have been observed in other groups, with the fertilization of non-nucleate ovules⁹ or the elimination of the maternal genome after fertilization¹⁰. In ants, both should spontaneously lead to males genetically identical to the sperm, as males are typically produced from haploid embryos through haplodiploidy¹¹.

Concern # 2:

The methods are complex and the writing in the manuscript is dense. In general, I had to work hard to understand which analysis was testing what. So, I think it will be easier if the Authors, step by step, associated the method used for each result and tell the Reader what they should see, as well as what the alternative pattern would mean. It is also important for the Authors to mention the level of statistical confidence they have in each findings, which this Reviewer could not easily ascertain from the data.

We appreciate this referee's feedback regarding the complexity and density of the methods section. Bioinformatics analyses indeed require detailed descriptions to ensure reproducibility, which can sometimes make the text challenging to follow.

To enhance clarity, we have revised the Methods section to better associate each method with its corresponding results. While the first subsections describe essential steps in the bioinformatics pipeline that are not tied to specific results, we have streamlined the explanations by adding introductory sentences that summarize the overall goal of each subsection. For seven key subsections, we have clearly outlined the expectations for each analysis, the methods used, and the statistical support for the findings, where applicable.

We hope that these changes will make the manuscript more accessible and help readers understand the connection between the methods and the results.

Concern # 3:

The Authors never really tell us why they selected the Messor species that they did. Given their previous study on this, why did they not use any of the species from the previous study? What would they have expected to find in these other species?

We understand the referee's question, as it highlights an aspect we did not detail in our effort to keep the manuscript concise. This study started as a population genetic study focusing solely on *M. ibericus*, which we suspected to hybridize with *M. structor*. Since the results were more and more confusing (hybrids without both parental species in the area), we expanded the sampling to the closest relatives of both *M. ibericus* and *M. structor*, which have been clearly identified from the literature as *M. ponticus*, *M. mcarthuri* and *M. muticus*¹². We now briefly mention this point in the Sampling subsection of the Methods section (line 279):

To better understand hybridization patterns between *M. ibericus* and *M. structor*, we gradually sampled individuals of both species, along with their respective closest relative species across Europe (*M. ponticus*, *M. muticus* and *M. mcarthuri*)¹²

The other species from our previous work are more distant phylogenetically than the five analyzed in the current work. Regarding our expectations:

- *M. barbarus* has been sufficiently sampled in our previous study (at least, in Europe), so we do not expect to find anything significantly different from what was already published, *i.e.* two genetic lineages of queens that systematically co-occur and hybridize to produce workers².

- Expectations are less clear regarding *M. ebeninus*, which is a more difficult species to sample from the Middle-East. This species may either have the same system as *M. barbarus* or the novel system of *M. ibericus*.

Concern # 4:

References 8 & 9 are not sufficient for this statement. Please find alternative or additional references for this statement.

We have added the following alternative references¹³⁻¹⁶ for more specific articles about the production of a distinct morph depending on seasonal conditions, population density or social caste.

Concern # 5:

It would be important to spell out the logic of why the authors think they are “1st generation hybrids”

We now specify in the main text that we used a method specifically designed to detect first-generation hybrids only (line 63):

We confirmed this hypothesis by conducting an analysis specifically designed to detect first-generation hybrids¹⁷, which identified all *M. ibericus* workers as such (see Methods, Table S1).

Following Concern #2, we also specify more clearly the logic of the approach in the Methods section.

We also detailed further the population structure analysis in the main text, specifying clearly that it is indicative of first-generation hybrids (line 74):

Finally, a population structure analysis¹⁸ on 5856 genes (44,191 variants) revealed that workers in *M. ibericus* colonies had virtually equal population ancestry proportions from *M. ibericus* and *M. structor* (averaging 0.49 and 0.51, respectively; Fig. 1 Table S1), which confirms further that they are first-generation hybrids.

Concern # 6:

They go back and forth between individual-level and colony-level (superorganism) perspectives and interpretations. This is confusing. I think it is better to state these in one place and talk about both alternatives.

The discussion has been refined extensively following the comments of the other referees, and we made our best to group all colony-level perspectives at the end of the discussion.

Referee #2

This manuscript is clear and well-written and uses an impressive array of methods to describe a fascinating example of “sexual parasitism”. Overall, I think this is very impressive work, and I wanted to be clear about that before moving on to present some critical comments.

We appreciate the reviewer's positive assessment of our manuscript, particularly the recognition of the fascinating nature of the described phenomenon. Before addressing the specific concerns, we would like to thank the reviewer for presenting several thought-provoking perspectives, which we found stimulating and valuable for clarifying our position on some semantic points worth being debated.

The central claim is the first known case of a "xenoparous" reproductive mode, where individuals "give birth to other species as part of their lifecycle." This novelty serves as the focus of the manuscript, so I will address it first. While it is indeed an exciting and novel finding, I do not find it as groundbreaking as the authors suggest.

We appreciate the reviewer's recognition of the novelty of our main claim, which indeed focuses on the discovery of “xenoparity” as a novel reproductive mode. Since this discovery is recognized as exciting and the concerns seem to revolve around our differing interpretations rather than a fundamental disagreement, we are hopeful that our clarifications will further enhance the reviewer's interest in this finding, even if only slightly.

For one, is egg-laying truly equivalent to giving birth? While this could be seen as a matter of semantics, I believe there is a significant distinction in the whole range of reproductive systems. Organisms exhibit a continuum of reproductive strategies, such as live birth, egg-laying, or broadcast spawning. Would the discovery of a broadcast spawner's egg developing into a different species still be considered groundbreaking? Or what if gametes from two different broadcast spawning species fused, resulting in an offspring of only one of the species?

The reviewer's point about the continuum of reproductive strategies is relevant. In the context of xenoparity, complex parental investment dedicated to another species (e.g. complete placental development in case of live-birth) does indeed appear more remarkable than no parental investment (e.g. simple gamete release in case of broadcast spawning). While we agree that xenoparous reproduction might seem less remarkable in a broadcast spawning context, ants exemplify the opposite extreme in this reproductive strategy continuum, even though they lay eggs rather than being viviparous. As social insects, ants represent one of the highest levels of parental investment among metazoans, exceeding even most viviparous species¹⁹. They exhibit extensive parental care from egg to adulthood, with the whole colony contributing to the rearing of the offspring. In the case of xenoparous reproduction, this means that allospecific individuals are produced by the queen and then raised to adulthood as siblings of the same family in spite of different species origin. Given the

significant parental investment of eusocial insects, the emergence of xenoparity is thus especially noteworthy, particularly when considering that colonial altruism evolutionarily relies on genetic proximity among same-species siblings.

What about a case where a zygote developed into a different species than the father who fertilised the egg? This phenomenon is already known (gynogenesis), and seems analogous, the main difference being that in one system the maternal gamete is parasitized, while in the other the paternal gamete is parasitized.

The referee is right when pointing out that gynogenesis (females parasitizing sperm) is analogous to androgenesis (males parasitizing ova). However, xenoparity is not simply one or the other. In gynogenesis, **sp1 females** use sperm of other species to produce new **sp1 females**. The interest is straightforward from the female's perspective, and as females control most of the reproductive process, gynogenesis evolved several times, including extreme cases of female-only species. Reversing this phenomenon is what is particularly exceptional about androgenesis: **sp1 females** producing **sp2 males** is not in the interest of females and is thus doomed to be rare and incidental, which likely is why there are no known male-only species to date. Now, what if **sp1 females** need **sp2 males** and thus do have an interest to produce them? This is xenoparity, which is the exception to the exceptions – and has been discovered in this work.

And as the authors themselves mention in the discussion, cross-species cloning via maternal gametes is already known too, although not in as complex a system as described here.

We indeed mention some cases of males parasitizing other species' ova, but their difference from xenoparity is not just a matter of complexity. It is again a fundamental difference in terms of **female interest**. In known cases of male parasitism, females have no interest in cloning another species by androgenesis. Those cases are thus by definition ruled out from the main novelty point of our study, as **there is no species needing to produce different species' individuals here**. In xenoparity, the females **have an interest and even absolutely need to clone the other species' males**. This is a fundamental difference that is not simply a difference in terms of complexity, as it leads to the unique situation where there is a requirement to propagate the genome of another species.

The point I am trying to convey is that there exists a wide range of unusual "sexually parasitic" reproductive modes, and while the system described in this study is striking, it does not seem entirely distinct from what is already known.

Sexual parasitism is indeed central in the history of our system, but xenoparity is by definition distinct, as it constitutes a further evolutionary step. There is indeed a wide range of either sperm or ova parasitism, **but the whole point of xenoparity is that**

it combines one into the other, resulting in a new reproductive unit (see the figure below): **sp1 females** “use” **sp2 male** sperm for worker production, **sp2 males** “use” **sp1 female** ova for cloning themselves. As **sp1 females** and **sp2 males** share the same colonies, both benefit from having their gametes used by the other, as both benefit from workers and male cloning (and even absolutely require each other's gametes in Southern Europe). By binding together sperm and ova parasitism into a single superorganism, xenoparity neutralized the sex war, as both species are trapped together into a self-sufficient unit of selection. We now include Fig. S18 (presented below for convenience), which clearly illustrates how “xenoparous reproduction” derives and differs from sexual parasitism.

Fig. S18. From sexual parasitisms to xenoparous reproduction.

Species that act as sexual parasites use another species' gametes to propagate their own genome⁷. Such sexual parasitism can be divided into two main categories illustrated at the top of this figure (a and b). Xenoparous reproduction (this study) is illustrated at the bottom of the figure (c).

a) Sperm parasitism. A female exploits the sperm of another species to reproduce. The most typical form of sperm parasitism is gynogenesis, where females reproduce asexually but require contact with sperm to initiate embryogenesis. After contact, the host male's genome is immediately excluded²⁰. In contrast, this exclusion is delayed to the next generation in hybridogenesis. In this alternate form of sperm parasitism, the male genome is present in the somatic cells of the parasite offspring but excluded during meiosis when germinal cells are produced²¹. In social insects, hybridogenesis refers to cases where the male genome is present in workers (*i.e.* somatic cells of a “superorganism”) but excluded from reproductive individuals (*i.e.* germinal cells of a “superorganism”)²². Such social hybridogenesis is common in the *Messor* genus² and is the sperm parasitism at the origin of xenoparous reproduction in *Messor ibericus* (see Fig. 3A).

b) Ova parasitism. A male exploits the ova of another species to reproduce. Typically referred to as male cloning, androgenesis is the flip side of gynogenesis, where offspring only carry the nuclear genome of their father. It is excessively rare compared with gynogenesis and typically occurs in a hermaphrodite context with exceptional inter-specific parasitism (*Cupressus dupreziana* conifers or *Corbicula* clams)⁸⁻¹⁰. To date, male clonality is restricted to intraspecific cases in ants²³⁻²⁶.

c) Xenoparous reproduction. Here, females need to produce males of the other species, as both species rely on each other's gametes. This combines elements of both sexual parasitism forms, but with a key difference: females are not victims of ova parasitism but require it, as it is the case for males regarding sperm parasitism. In the sole case known so far (this study) **sp1 females** “use” **sp2 male** sperm for worker production, while **sp2 males** “use” **sp1 female** ova for cloning. Since **sp1 females** and **sp2 males** share the same colonies, both species benefit from having their gametes exploited by the other, as both require workers and males. By integrating sperm and ova parasitism into a single superorganism, such a “sexual domestication” neutralizes the sex war, as both species are trapped into a self-sufficient unit of selection. As it results in a cohesive reproductive unit of two species, evolution toward xenoparity can be qualified as an evolutionary transition in individuality^{27,28}.

My understanding is that the crucial element here is that a fertilized gamete from a mother can produce progeny of a different species, regardless of the exact method of offspring delivery, and that this different species is necessary for maintenance of the first species. This concept closely mirrors cases like gynogenesis, where a gamete from a father fuses with an egg to produce progeny of a different species, independent of the delivery process. While I do not think the central thesis is wrong, I do think it should be more fairly integrated with current understanding of such parasitic reproductive systems.

We thank the referee for making us realise that the manuscript was not clear enough about the extent of the difference between xenoparity and the various types of parasitic reproductive systems. We hope that our last answer clarified the fact that xenoparity is an evolutionary step distinct from sexual parasitism (to which gynogenesis belongs) and that it is not simply a matter of the delivery process (as both gynogenesis and androgenesis fail to meet the main novelty claim of the article). To become xenoparous, gynogenetic females (or sperm parasites in general) need to acquire the ability to produce by themselves their host species males, thus trapping them into their lifecycle. To date, such a “sexual domestication” phenomenon is unique and reported for the first time in this work.

We have modified the discussion in this regard by referring to the above figure as supplementary Figure S18 to better integrate our system among other parasitic reproductive systems (line 257):

By depending on each other's gametes, both species have intertwined their lifecycles, evolving from sexual parasitism⁷ to sexual codependency (Fig. S18).

The discussion is quite speculative and, in some respects, exaggerates the meaning or novelty of the results. Here the authors briefly discuss androgenesis, but quite superficially and it is not clear what the implication is. As I understand it, the message is that cross-species cloning via females is known in other species too, but it is not obligate and is thus less striking than the system described here. This is of course true to an extent, but nevertheless, it does decrease the novelty of the current results in a similar way to gynogenesis discussed above.

We thank the referee for pointing out that our brief discussion about androgenesis was more confusing than informative, as we did not want to imply that xenoparity is just “obligatory androgenesis”. The referee is correct when pointing out that our system is (among other things) the sole obligatory instance of cross-species cloning, but **the main novelty point is not a matter of obligatory/facultative system**. Among the known cases of androgenesis, even if one would imagine males that obligatorily need to parasitize other species' ova, it would still be fundamentally different from xenoparity, as it would be at the expense of parasitised females. As explained in a previous comment, xenoparity is exceptional in the sense that **females are not victims of cloning other species' males, but require it**.

To avoid this confusion in the discussion, we have rephrased the first paragraph to be clear about the fact that xenoparous reproduction is not just obligatory androgenesis (line 235).

To date, females needing to clone members of another species have not been observed. Although cross-species cloning has been reported in hermaphrodite conifers and clams⁸, these are instances of male parasites occasionally hijacking other species' eggs. In such cases, producing males of another species is not in the interest of females, as they are incidental victims of parasitism. This contrasts with the system reported here, where producing other species' male is not an accident, but a female lifecycle requirement. We suggest defining such females as xenoparous, meaning they need to produce individuals of another species as part of their lifecycle. This reveals the evolution of xenoparity (xeno- meaning “foreign, strange, different” and -parity meaning “produce, bring forth, give birth”), which is the need to propagate another species' genome via its own eggs.

The authors then describe xenoparity as a symbiotic mutualism from an evolutionary perspective, and the males in the system as a perfected form of male parasites. Clonal males are discussed as organelles at the superorganism level, and the manuscript closes by stating that this xenoparous colonial lifeform challenges usual boundaries of organismic complexity. All of this is interesting, but it is not really backed up by the results. That the system has evolved as a symbiotic mutualism seems quite uncertain to me, or that the males can really be considered perfected parasites. If I was to speculate, I would guess the most likely explanation for such a system is a type of evolutionary trap where 'the best of a bad job' (once a dependence on males of a different species has arisen perhaps partly by chance events) is maintained. It is not clear to me how such a system might be beneficial to the host species, over the 'typical' way of making workers and males. Are the males perfected parasites? It seems more like they have been co-opted into the life-cycle of the host. Do they have the possibility of their spreading their genes outside of this domesticated situation? If not, is it again more like an evolutionary trap than perfected parasitism?

We thank the referee for sharing all these interesting thoughts. Before addressing the criticisms, we would like to clarify that we fully agree with the reviewer's perspective regarding the "best of a bad job" and evolutionary trap scenario. This interpretation aligns with our view on the origin of the system. For example, Figure 3 and its corresponding result section imply that clonal *M. structor* males have been co-opted / domesticated into the lifecycle of a host that initially depends on their sperm, which can be indeed viewed as a form of evolutionary trap. We thank the referee for making us realize that this point was not clear enough. We have now corrected this shortcoming in various parts of the manuscript.

First, as stated in the response of a previous comment, we refer to the figure above (new Fig S18) that illustrates how clonal males are evolutionarily trapped within the *M. ibericus* lifecycle.

Second, we now mention the analogy with domestication right from the result section, using the referee's well-chosen term "co-opted" (line 188):

This pathway appears analogous to domestication⁵, as *M. ibericus* co-opted *M. structor* males into its lifecycle, maintaining them as a clonal lineage rather than exploiting them from the wild.

Finally, we have added a new paragraph to the discussion, where we mention previous works supporting the idea that evolution towards obligatory sperm parasitism (the ancestral state of xenoparity, see Fig. 3A) is the result of a "best of a bad job" scenario rather than an adaptive advantage compared with the typical ant reproductive system³ (line 245):

The uncovered reproductive mode appears to result from a “best of a bad job” scenario. Like several other harvester ant species, *M. ibericus* first became locked into obligate sperm parasitism^{1,2} (Fig. 3a), a situation where they lost the ability to produce workers by themselves due to epistatic incompatibilities^{4,29} or selfish caste-biasing genotypes³. This dependence on males from another species is sub-optimal for queens, as it requires them to mate with two different male partners and restricts their colonies to the geographic range of their host. By producing the required species’ males in their own colonies (Fig. 3b), *M. ibericus* has gained a clear advantage, as it maintains the “best of a bad job” situation while minimizing the inherent constraints.

That being said, this scenario is not incompatible with our points about "**symbiotic mutualism**" and "**perfected parasites**". These terms were not meant to suggest alternative evolutionary pathways, but rather to compare the system with typical biological interactions and contextualize it within the parasitism-mutualism continuum. Here are some clarifications regarding these two points.

1) Regarding the point about "**symbiotic mutualism**," the referee prefers to describe the clonal males as being in a "domesticated situation". We fully agree with this precision, as domestication is a specific form of mutualism. For example, mutualism and domestication are often used interchangeably in the case of leaf-cutting ants cultivating a mushroom³⁰. Both terms are also analogous when considering the evolutionary concept of domestication, as defined by: "*A broad biological definition of domestication is that it is a coevolutionary process arising from a **mutualism**, in which one species (the domesticator) constructs an environment where it actively manages both the survival and reproduction of another species (the domesticate) in order to provide the former with resources and/or services.*"⁵

Based on this definition, qualifying the clonal males as being in a "domesticated situation" acknowledges the mutualistic nature of their relationship. This definition applies to human-driven domestication examples, but also matches particularly well the case of our ants, as *M. ibericus* construct an environment (the nest) where they actively manage both the survival and reproduction of *M. structor* males (as they produce and take care of them) in order to provide a resource/service (sperm for producing workers). Note that even in a “domesticated situation” or in an “evolutionary trap”, the situation is highly beneficial to clonal males: in the end, an identical copy of their genome is spread to the next generation, with all the cost in terms of ova, nest or resources being paid by *M. ibericus*. That being said, we understand that qualifying domestication as a mutualism is not obvious at first read, and that such semantic clarifications might not be appropriate in a discussion expected to be clear and concise. We have thus replaced the term “symbiotic mutualism” by “sexual codependency”, which is both more precise and less confusing at first read.

2) Similarly, our comparison of clonal males with "**perfected parasites**" may have been misinterpreted. This term is grounded in the widely accepted notion that

parasites causing excessive harm to their hosts are poorly adapted, as they risk driving both the host and themselves to extinction³¹. Ideally, a “perfect” parasite should minimize harm to its host or, even better, chaining its host to the interaction by becoming indispensable to it, as seen in the many cases of symbiotic mutualism that have evolved from parasitism^{32,33}.

A clonal male-only lineage that is indispensable to its female host exemplifies such a “perfect parasite” concept in the context of typical male parasitism. Again, we did not intend to imply that clonal males are not in an “evolutionary trap”. Rather, we emphasize that being evolutionarily trapped within its female host lifecycle (see Fig. 17 above) is the ideal situation for reproducing clonally via other species’ ova. We have rephrased this part to clarify this point (line 254):

While trapped in the lifecycle of a species exploiting their sperm, clonal males propagate their genome through the reproductive efforts and parental care of *M. ibericus*. In a sense, clonal males can be viewed as a perfected form of male parasites, as they are essential to their female hosts but reproduce at the expense of their ova. By depending on each other’s gametes, both species have intertwined their lifecycles, evolving from sexual parasitism⁷ to sexual codependency (Fig. S18).

Finally, I don’t think the system really challenges usual boundaries of organismic complexity. It does have a novel feature, but there are many other species with complex superorganism colonies that surpass the complexity of this one in some other way.

Complexity can indeed apply to various criteria, and we did not want to imply that an *M. ibericus* colony is the most complex in all of them. When mentioning specifically “organismic complexity” and citing the article “*Major evolutionary transitions in individuality*”²⁸, we only wanted to refer to complexity in terms of individuality within “*The major transitions in evolution*” framework²⁷. In this seminal work, Maynard Smith and Szathmàry argue that some evolutionary transitions redefine what is an individual organism as the unit of selection. A famous example is the transition from unicellular to multicellular organisms, which further evolved into superorganisms.

West et al. (2015) expanded the concept of major evolutionary transition to obligate mutualism, where “two previously independent species have become interdependent into an integrated entity”²⁸. Among superorganisms, fungus-growing ant colonies are exemplified as such a major transition, as it is the case for ant-bacterial symbiosis in a more recent article³⁴. These standpoints are certainly debatable, but given that bacterial and fungus domestication are considered as such in highly influential articles, we believe that a unique case of “male domestication” should be at least briefly mentioned through this perspective. This seems all the more relevant when considering the extent of the sexual and social integration of clonal males within *M.*

ibericus colonies. Contrary to fungus/bacterial domestication, such a “male domestication” involves direct sexual interdependence for reproduction, as both partners rely on each other’s gametes. Produced by the same queens, they interact as siblings of the same superorganism in spite of a different species origin, which is akin to different species’ cells in a multicellular organism. This contrasts with fungus-growing ants and ant-bacterial symbiosis, where fungi and bacteria do not engage in sexual and social interactions with the ants and thus cannot be considered full-fledged members of the colony.

Because superorganisms are by definition composed of closely related, same-species individuals, we believe that such a “two-species superorganism” certainly blurs a little bit more “usual boundaries of organismic complexity”. Given that this concluding remark is not a critical claim but a thought provoking opening, we did not elaborate extensively on this point in the manuscript. We therefore have added citations and refined the last sentence to better contextualize this perspective:

Besides revealing a reproductive mode **under which** one species needs to clone another, such a **“two-species superorganism”** challenges the usual boundaries of organismic complexity **regarding evolutionary transitions in individuality**^{27,28,34}

Referee #3

Summary of Key Results; Originality and Significance: In their manuscript, Juvé et al. lay out a compelling case for their novel discovery of an ant that produces both “normal” offspring and heterospecific males. They show that all *Messor ibericus* workers are F1 hybrids (with *M. structor* fathers) and, in allopatric regions where heterospecific males are not otherwise present, *M. ibericus* queens produce a clonal lineage of *M. structor*-derived males. This stunning finding is laid out in a clear and convincing manner in the article. I am convinced that the main result is valid, but I suggest that the authors take one additional step to demonstrate that the clonal male lineage produced by *M. ibericus* queens is still functionally *M. structor*, as the authors claim. In addition to this major suggestion, I have some minor comments and ideas.

Data and methodology: The methods appear to be appropriate, and are clearly explained and documented.

Conclusions: The conclusions are generally robust and well supported.

We thank the referee for all the nice comments about our work.

Suggested improvements:

Major suggestion:

This article is framed as an instance of what the authors term “xenoparity”, wherein an individual of one species produces offspring of another species. The authors systematically demonstrate that single *M. ibericus* queens can produce two types of males: those with *M. ibericus* mitochondrial and nuclear genomes and those with *M. ibericus* mitochondrial genomes and *M. structor*-derived nuclear genomes. The latter clonal males form a sister clade to the so-called “wild type” *M. structor* males (and queens/workers and are morphologically distinct from both *M. ibericus* males and *M. structor* wild type males). Since these males father *M. ibericus* (hybrid) workers, they clearly retain viability. However, in order to demonstrate that these males are indeed *M. structor*, as the authors argue, it is necessary to demonstrate that they can still mate with *M. structor* queens and produce viable offspring (at least in the lab, if not in the field). To me, this seems like a final piece of the puzzle that is needed in order to show that these males are truly still *M. structor* and not a distinct lineage of hybrid origin in their own right. To be clear, even without this addition, I still think that the results of this study are very exciting. However, if the authors do not demonstrate that the *M. ibericus*-produced heterospecific males can still successfully mate with *M. structor* queens, I would suggest softening the language (perhaps by referring to these males as having *M. structor*-derived nuclear genomes, as I do above, or similar).

We appreciate the referee’s suggestion, as we have extensively debated ourselves the taxonomic characterisation that the clonal males should have. As suggested by the referee, one could argue that they still belong to *M. structor* (as we did) or that they deserve to be classified as distinct in their own right. Our decision to maintain the current classification was to avoid a potentially far-fetched claim, **as stating that clonal males are not *M. structor* would imply that we uncovered the first male-only species ever found**. By choosing to strictly stick to the established taxonomy in this group, we thus made a conservative choice that does not take the risk of overselling our results. We therefore agree with the reviewer’s point that this conservative choice might have been too conservative, but want to make clear that changing our stance would not soften our discovery but would rather double the

original claim (one species produce males from two species) with another one (one of the produced species is male-only).

To explore further this interesting question, we first followed the reviewer's suggestion by searching for offsprings resulting from matings between clonal males and *M. structor* queens. Such an analysis is more challenging than screening for clear interspecific hybrids, as the divergence level of the two potential parents is low (average of 0.0015 among loci) compared with that of *M. ibericus* and *M. structor* (average of 0.0052 among loci). To address this challenge, we enhanced our hybrid detection method¹⁷ by employing an *in silico* hybridization approach to adjust for such low heterozygosity cases (see details in Supplementary Information section 4). Applying this approach on 45 *M. structor* female genomes, we did not find such hybrids, as the heterozygosity and γ hybrid index of *M. structor* genomes are significantly lower than the expected values of a wild/clonal hybrid (Fig S13). This reinforces the view that clonal males are essentially trapped in a domesticated situation, as it confirms that they rarely reproduce sexually with *M. structor* females. Does this mean that clonal males constitute a distinct species? Not necessarily, as both lineages might just encounter each other extremely rarely, particularly when considering that clonal males are born in *M. ibericus* colonies while *M. structor* females generally mate within their native nests³⁵. Seeking rare hybrids in field or lab settings is also inherently limited, as these approaches provide insights only over short timescales, whereas the speciation process operates over many generations. For example, finding rare hybrids would not necessarily indicate that F2 offspring are sufficiently fertile to enable long-term gene exchanges between the two lineages. Fortunately, these limitations can be circumvented by taking advantage of our large comparative population genomics dataset, enabling us to reliably assess long-term hybridization histories and thereby accurately delimit species.

To specifically address this question, we conducted seven supplementary analyses detailed in Supplementary Information section 4 and Fig. S14-16. In summary, **these analyses consistently support the view that clonal males should be considered part of *M. structor***, as two different species delimitation methods grouped clonal males as the same species as wild-type lineages (Fig. S14). Further supporting the idea, clonal males and *M. structor* (wild-type) do not always split in clear monophyletic clades from in-depth phylogenomic analyses (Fig. S14a). Additionally, a population structure analysis indicated that clonal males are not of hybrid origin, as they share the same population of origin as other *M. structor* (Fig. S14b). Moreover, the F_{st} statistic for population differentiation between clonal and wild-type lineages is similar to typical intraspecific comparisons compared to interspecific ones (Fig. S15). Finally, the genetic divergence and gene flow exchanges between clonal and wild-type lineages are comparable with those observed at the intraspecific level in other *Messor* species (Fig. S16).

Taken together, these results reinforce our initial decision to consider clonal males as not sufficiently genetically distinct from *M. structor* to warrant classification as a separate species. In the meantime, empirical evidence indicates that they are currently mostly prevented from mating with their wild counterparts, which strengthens the view that they should be characterized as a domesticated lineage of *M. structor*. The relevance of considering clonal males as a domesticated lineage is underscored by their low genetic differentiation yet high morphological divergence, a pattern that mirrors other cases of domestication (e.g. dogs vs wolves, pigs vs boars...) ³⁶. As a result, we now more explicitly characterize clonal males as a *M. structor* “domesticated lineage” in the main text. All supplementary figures related to this part are provided below for convenience, while more details about methods and results are available in Supplementary Information section 4. Related changes in the main text (line 205) are readable here:

To assess whether clonal males can escape their “domesticated” situation by mating with their wild female counterparts, we conducted a detailed analysis on 45 *M. structor* genomes to detect potential hybrids (Supplementary Information section 4). Our findings confirmed that such events are currently non-existent or extremely rare, as we did not identify any hybrid between clonal and wild-type lineages (Fig. S13). Similarly to typical cases of domestication, this raises the question of whether recent genetic isolation from wild populations warrants a different species classification ³⁷. Further analyses therefore support that clonal males still belong to *M. structor*, as phylogenetic conflict (Fig. S14a), population genetic structure (Fig. S14b), species delimitation inferences (Fig. S14d-e), low F_{st} fixation index (Fig. S15), low genetic divergence (Fig. S16a), and high historical gene flow (Fig. S16b) are all consistent to support clonal and wild-type lineages as part of the same species (see details in Supplementary Information section 4). Taken together, these results further support the idea that clonal males should be characterized as a domesticated lineage of *M. structor*. All in all, this means that *M. ibericus* females interact with up to three males that are morphologically and genetically distinct (*M. ibericus*, “domesticated” *M. structor* and “wild” *M. structor* males; Fig. S12), laying two of them (Fig. 2) and mating with the three (Fig. 3).

Fig. S13

Fig. S13. SNP heterozygosity and γ hybrid index distribution of 45 *M. structor* genomes. The expected value in case of Clonal/Wild-type hybrid is in red and has been inferred from *in silico* hybridization using one haploid male genome from each lineage.

Fig. S14

Fig. S14. Phylogenetic tree and species delimitation. **a**, Phylogenetic tree of top 6 representative individuals per species (inferred from a 11,102,444 bp supermatrix with IQ-TREE). Support values supporting the monophyly of each species are written in the corresponding color. First number is the ultrafast bootstrap value, second is the Site Concordance Factors (sCF). **b**, Population ancestry proportion from population structure analysis. **c**, Species delimitation using *tr2*. Same color bar refers to the same species assignment by the method. **d**, Species delimitation using *SODA*. Same color bar refers to the same assignment by the method.

Fig. S15

Fig. S15. F_{st} Variation across intraspecific and interspecific pairs. Pairs are ranked according to their median F_{st} values. For intraspecific comparisons, we divided each species into two populations based on geographical origin. ***M. structor clonal* west** (SH04-18-reseq, SH19-01) vs ***M. structor clonal* east** (SH04-02-reseq, SH02-24-reseq); ***M. ibericus* west** (SH09-55, SH19-06) vs ***M. ibericus* east** (SH10-25, SH12-04); ***M. muticus* west** (Y15472-1, Y15473-1) vs ***M. muticus* east** (Y15479-1, Y15452-1); ***M. ponticus* north** (SH03-21, RDNIPQ) vs ***M. ponticus* south** (Y14753-1, SH01-17); ***M. structor wild-type* west** (Y16627-1, Y15267-1, Y15268-1) vs ***M. structor wild-type* east** (Y14750-1, RBUDAW, Y14582-1); ***M. mcarthuri* west** (LGR15Q, Y16370-1) vs ***M. mcarthuri* east** (Y15243-1, Y15241-1).

Fig. S16

Fig. S16. Divergence and gene flow between wild-type and clonal lineage compared with intra- and interspecific comparisons. a, Genetic divergence between all wild-type vs. clonal pairs ($n=36$) compared with intraspecific pairs ($n=90$) and interspecific pairs ($n=504$). Dark grey background delimits the divergence range of interspecific pairs, white background the intraspecific one, light grey the divergence range where both overlap. **b,** Gene flow proportion inference within all wild/wild/clonal triplets ($n=90$) compared with intraspecific triplets (spA/spA/spA, $n=100$) and interspecific triplets of 2 species (spA/spA/spB, $n=1965$). Dark grey background delimits the gene flow proportion range of interspecific triplets, white background the intraspecific one, light grey the overlap.

Minor comments and suggestions:
Line 26: individuals -> offspring?

Change done.

Line 39: Such a complexity -> Such complexity; same line, consider removing “need to additionally”

We have implemented all suggested modifications.

Line 42: contrasts -> variation? Or distinctiveness?; same line, consider omitting “required to be”

We have changed “contrasts” by “variation” and removed “required to be”.

Line 46: omit “yet”?

Change done.

Line 49: consider removing “exclusively”

Change done.

Line 51: I realize that this “symbiosis” idea was explained later, but from the context provided so far, it is not apparent that *M. structor* requires the gamete of *M. ibericus* (the *M. ibericus*-produced heterospecific males require *ibericus* ova, but the *M. structor* species does not require a heterospecific gamete to persist”. This sentence really confused me on the first read through.

Thanks for pointing this out; it is indeed confusing to read about “symbiosis” at this point. We have changed the end of the sentence to now read:

“which suggests **a domestication-like process for exploiting another species’ gametes**”

Line 58: consider rewording or at least moving or removing “females”- this formulation was confusing.

We had used “females” to imply that only females are expected to feature similar heterozygosity, as they are diploids while males are haploids. The formulation is indeed very confusing in the end – thank you for pointing this out –, and so we have replaced females by “diploid individuals”.

Lines 58-61: I noticed that the total sample size presented here did not match the 390 individuals mentioned above.

This is normal; these comparisons only include females, as heterozygosity can only be measured on diploid individuals. The 390 individuals mentioned above also include males, which are analysed when not computing heterozygosity.

Line 78: other -> another

Change done.

Line 80: comes as -> is

Change done.

Line 123: of -> for

Change done.

Line 145: omit the first instance of “two”?

Change done.

Lines 185-188: This is an interesting idea, but I'm not sure that the authors can conclude that the morphological difference resulted from the loss of genetic diversity. Couldn't there also be an interaction between genetic markers in the nuclear and mitochondrial genomes, plasticity due to different rearing conditions, or any number of other causes of this difference? The authors mention some alternative possibilities later in the paragraph, but I found this sentence to be a little misleading.

We agree that the formulation might imply a causation that we did not want to suggest here. To remove ambiguity, we have changed the sentence to now read:

Interestingly, clonal males also differ morphologically

We have also integrated the well-taken hypothesis about a difference that may be due to plasticity due to different rearing conditions (line 203).

Line 189: “compared to the others” – which others do the authors mean?

We now specify:

“compared with the wild-type and *M. ibericus* males”.

Line 212: awkward wording (In particular, “the need to clone” seems a little strange)

We have replaced this sentence by:

To date, females needing to clone members of another species have not been observed.

Line 221: appears as -> appears to be a

Change done.

Lines 231-244: Very cool parallels!

We particularly appreciate the positive feedback of the reviewer about the final part of our discussion, as this part is dedicated to the significance of the discovery.

Lines 401-405: This phasing approach seems a little simplistic and likely to lead to incorrect inference, as described here: "Based on sampling and previous results (mitochondrial phylogeny, Fig. S1), we know that *M. ibericus* is the maternal species of hybrid workers, so that the alleles that do not match a *M. ibericus* genome belong to the paternal species. We exploited this by writing a python script comparing each SNP of each hybrid worker with variants of a reference maternal genome (*M. ibericus* queen genome with the highest coverage, SH19-06)." Overall, this approach probably gets you close enough, but the statement that I italicized seems problematic to me if there is substantial genetic variation across the sampled range of *M. ibericus*.

The reviewer is entirely right; such an approach is expected to bias the paternal genome reconstruction in case of substantial genetic diversity within the maternal species. However, we used this approach precisely because the genetic diversity of *M. ibericus* is exceptionally low. As reported, in table S4, the measured synonymous diversity π_s is of 0.00045. For reference, this is below the genetic diversity of the giant Galapagos tortoise (0.0016) and is the lowest among 76 animal species examined in a comparative study¹⁹. We now clarify this important point, as this approach is particularly well-suited to our case but may not be suitable for other species. Here are our modifications:

Previous results indicated that *M. ibericus* is the maternal species of all hybrid workers, as hybrids are only found in *M. ibericus* queen colonies (see also Fig. 1, Fig. S1). Consequently, alleles of hybrid individuals that do not match *M. ibericus* are expected to belong to the paternal species. Given the exceptionally low genetic diversity of *M. ibericus* (Table S4, π_s of 0.00045), the risk of confusing intraspecific polymorphism with paternal alleles is minimized. We exploited this specificity by writing a python script comparing each SNP of each hybrid worker with variants of a reference maternal genome (*M. ibericus* queen genome with the highest coverage, SH19-06).

References: To the extent that I am familiar with this subfield, it appears that previous work is cited appropriately.

Clarity and context: I found the article to be written in a clear and compelling way.

We thank the referee for these positive final comments regarding the references and clarity of the article.

Referee #4

Review of manuscript “One mother for two species: obligate cross-species cloning in ants” by Juve et al.

The research team reports an extraordinary type of reproduction in ants. Apparently, the queen produces males of two different species. And then has to mate with males of these two species in order to produce a functional colony because ‘hybrid’ workers are necessary to success.

I found this work very interesting. Several ant species have been found to engage in unusual reproductive mechanisms in the past 20 years. But this new proposed mechanism is quite amazing and unique.

We appreciate the reviewer's positive feedback on our manuscript. We are particularly pleased that the reviewer recognizes the *M. ibericus* system as “unique” and “amazing”, especially given their evident familiarity with the diverse and unusual reproductive systems uncovered in ants. This perspective further emphasizes the significance and novelty of this system, and we are glad that our work has been received with such interest.

However, the one question I had was about mechanism. What exactly is happening from a developmental and reproductive standpoint to allow this to occur. Queens are apparently not ‘hybrid’ so are queens polyandrous?

We confirm that queens are absolutely not hybrids, as it can be more clearly seen in the new Fig. S14b where *M. ibericus* queens clearly share a single population ancestry. We also confirm that *M. ibericus* queens are polyandrous, as they store sperm of *M. ibericus* and *M. structor* males to produce the queen and worker caste. We now highlight better this point in the main text by explicitly mentioning that queens are polyandrous at line 147:

Consistent with this explanation, ***M. ibericus* are polyandrous and mate with both species males**, as we retrieved spermatozoa of both *M. ibericus* and *M. structor* when sequencing the spermatheca content of a queen that gave birth to both species

And then the male is essentially cloning itself through its sperm as the authors suggest? I wasn't sure exactly how this system could occur. But I did feel that some greater insight into mechanism would be important to having a complete story about this extraordinary system.

M. ibericus queen genomes being non-hybrid, the only possible explanation for our observations is that their *M. structor* sons are direct copies of the corresponding sperm that they store in their spermatheca. Male cloning as genetic copies of spermatozooids is a phenomenon coined as androgenesis in the literature, whereby maternal genetic material is eliminated before or after fertilization. In response to Reviewer 1 and this comment, we now provide greater insight into the known mechanisms behind this process at line 139:

This observation points to androgenesis (*i.e.* male clonality), where a male provides the sole source of nuclear genetic material for the embryo⁸. Embryos devoid of maternal DNA have been observed in other groups, with the fertilization of non-nucleate ovules⁹ or the elimination of the maternal genome after fertilization¹⁰. In ants, both should spontaneously lead to males genetically identical to the sperm, as males are typically produced from haploid embryos through haplodiploidy¹¹.

References

1. Umphrey, G. J. Sperm parasitism in ants: selection for interspecific mating and hybridization. *Ecology* **87**, 2148–2159 (2006).
2. Romiguier, J., Fournier, A., Yek, S. H. & Keller, L. Convergent evolution of social hybridogenesis in Messor harvester ants. *Mol Ecol* **26**, 1108–1117 (2017).
3. Weyna, A., Romiguier, J. & Mullon, C. Hybridization enables the fixation of selfish queen genotypes in eusocial colonies. *Evol Lett* **5**, 582–594 (2021).
4. Helms Cahan, S. & Keller, L. Complex hybrid origin of genetic caste determination in harvester ants. *Nature* **424**, 306–309 (2003).
5. Purugganan, M. D. What is domestication? *Trends Ecol. Evol.* **37**, 663–671 (2022).
6. Rabeling, C. & Kronauer, D. J. C. Thelytokous parthenogenesis in eusocial Hymenoptera. *Annu Rev Entomol* **58**, 273–292 (2013).
7. Lehtonen, J., Schmidt, D. J., Heubel, K. & Kokko, H. Evolutionary and ecological implications of sexual parasitism. *Trends Ecol. Evol.* **28**, 297–306 (2013).
8. Schwander, T. & Oldroyd, B. P. Androgenesis: where males hijack eggs to clone themselves. *Philos. Trans. R. Soc. Lond. B Biol. Sci.* **371**, (2016).
9. Pichot, C., Borrut, A. & El Maâtaoui, M. Unexpected DNA content in the endosperm of *Cupressus dupreziana* A. Camus seeds and its implications in the reproductive process. *Sex. Plant Reprod.* **11**, 148–152 (1998).
10. Komaru, A., Ookubo, K. & Kiyomoto, M. All meiotic chromosomes and both centrosomes at spindle pole in the zygotes discarded as two polar bodies in clam *Corbicula leana*: unusual polar body formation observed by antitubulin immunofluorescence. *Dev Genes Evol* **210**, 263–269 (2000).
11. Heimpel, G. E. & de Boer, J. G. Sex determination in the hymenoptera. *Annu. Rev. Entomol.* **53**, 209–230 (2008).
12. Steiner, F. M. *et al.* Turning one into five: Integrative taxonomy uncovers complex evolution of cryptic species in the harvester ant Messor ‘structor’. *Mol. Phylogenet. Evol.* **127**, 387–404 (2018).
13. Simpson, S. J., Sword, G. A. & Lo, N. Polyphenism in insects. *Curr. Biol.* **22**, 352 (2012).

14. Schwander, T., Lo, N., Beekman, M., Oldroyd, B. P. & Keller, L. Nature versus nurture in social insect caste differentiation. *Trends Ecol. Evol.* **25**, 275–282 (2010).
15. Shapiro, A. M. Seasonal Polyphenism. in *Evolutionary Biology* 259–333 (Springer US, Boston, MA, 1976).
16. Applebaum, S. W. & Heifetz, Y. Density-dependent physiological phase in insects. *Annu. Rev. Entomol.* **44**, 317–341 (1999).
17. Weyna, A., Bourouina, L., Galtier, N. & Romiguier, J. Detection of F1 Hybrids from Single-genome Data Reveals Frequent Hybridization in Hymenoptera and Particularly Ants. *Mol Biol Evol* **39**, (2022).
18. Raj, A., Stephens, M. & Pritchard, J. K. fastSTRUCTURE: variational inference of population structure in large SNP data sets. *Genetics* **197**, 573–589 (2014).
19. Romiguier, J. *et al.* Comparative population genomics in animals uncovers the determinants of genetic diversity. *Nature* **515**, 261–263 (2014).
20. Schlupp, I. The evolutionary ecology of gynogenesis. *Annu. Rev. Ecol. Evol. Syst.* **36**, 399–417 (2005).
21. Lavanchy, G. & Schwander, T. Hybridogenesis. *Curr. Biol.* **29**, 539 (2019).
22. Leniaud, L., Darras, H., Boulay, R. & Aron, S. Social hybridogenesis in the clonal ant *Cataglyphis hispanica*. *Curr. Biol.* **22**, 1188–1193 (2012).
23. Ohkawara, K., Nakayama, M., Satoh, A., Trindl, A. & Heinze, J. Clonal reproduction and genetic caste differences in a queen-polymorphic ant, *Vollenhovia emeryi*. *Biol. Lett.* **2**, 359–363 (2006).
24. Pearcy, M., Goodisman, M. A. D. & Keller, L. Sib mating without inbreeding in the longhorn crazy ant. *Proc. Biol. Sci.* **278**, 2677–2681 (2011).
25. Okita, I. & Tsuchida, K. Clonal reproduction with androgenesis and somatic recombination: the case of the ant *Cardiocondyla kagutsuchi*. *Naturwissenschaften* **103**, 22 (2016).
26. Rey, O., Facon, B., Foucaud, J., Loiseau, A. & Estoup, A. Androgenesis is a maternal trait in the invasive ant *Wasmannia auropunctata*. *Proc. Biol. Sci.* **280**, 20131181 (2013).
27. Maynard-Smith, J. & Szathmary, E. *The Major Transitions in Evolution*. vol. 49 1302 (OUP Oxford, 1997).
28. West, S. A., Fisher, R. M., Gardner, A. & Kiers, E. T. Major evolutionary transitions in individuality. *Proc. Natl. Acad. Sci. U. S. A.* **112**, 10112–10119 (2015).

29. Anderson, K. E. *et al.* Distribution and evolution of genetic caste determination in Pogonomyrmex seed-harvester ants. *Ecology* **87**, 2171–2184 (2006).
30. Schultz, T. R. *et al.* The coevolution of fungus-ant agriculture. *Science* **386**, 105–110 (2024).
31. Lenski, R. E. & May, R. M. The evolution of virulence in parasites and pathogens: reconciliation between two competing hypotheses. *J. Theor. Biol.* **169**, 253–265 (1994).
32. Ewald, P. W. Transmission modes and evolution of the parasitism-mutualism continuum. *Ann. N. Y. Acad. Sci.* **503**, 295–306 (1987).
33. Zug, R. & Hammerstein, P. Bad guys turned nice? A critical assessment of Wolbachia mutualisms in arthropod hosts. *Biol. Rev. Camb. Philos. Soc.* **90**, 89–111 (2015).
34. Rafiqi, A. M., Rajakumar, A. & Abouheif, E. Origin and elaboration of a major evolutionary transition in individuality. *Nature* **585**, 239–244 (2020).
35. Schlick-Steiner, B. C. *et al.* More than one species of Messor harvester ants (Hymenoptera: Formicidae) in Central Europe. *Eur. J. Entomol.* **103**, 469–476 (2006).
36. Andersson, L., & Purugganan, M. Molecular genetic variation of animals and plants under domestication. *Proc. Natl. Acad. Sci, U. S. A.* **119**, e2122150119 (2022).
37. Gentry, A., Clutton-Brock, J. & Groves, C. P. The naming of wild animal species and their domestic derivatives. *J. Archaeol. Sci.* **31**, 645–651 (2004).

Response to Reviewer' comments for manuscript 2024-11-23818B-Z in Nature

One mother for two species: obligate cross-species cloning in ants

Note: To enhance the readability of this response letter, all the reviewers' comments are typeset in blue boxes. Rephrased or added sentences in the main manuscript are typeset in red.

Referee #1 (Remarks to the Author):

The revised version of "One mother for two species: obligate cross-species cloning in ants" by Juve et al. has much improved. The Authors have addressed all my concerns from the previous round of review. I have just have three points to help improve the manuscript before publication:

We appreciate the reviewer's positive feedback on our revision and are grateful for the suggestions provided to help improve the manuscript before publication.

Point # 1:

The reader would benefit from more background information about the selective advantage of maintaining hybrid workers for the colony. Without this key piece of information, it is hard to understand why they would be maintained in the colony. For example, on Lines 168-169, the Authors state there are many well-described examples of queens using other species' sperm to produce their workers (and suggest this is the ancestral state of their study system). They cite many references describing hybrid workers, but to someone not familiar with this literature it is a very surprising finding in itself, and it would be very helpful to briefly mention why such a system might evolve (why use another species' sperm vs your own?).

This is a good point to improve the readability for a large audience. As suggested, we now briefly mention the evolutionary reasons mentioned in the literature about hybridization for worker production right after the lines mentioned by the referee.

This strategy may have been selected either to benefit from potential worker hybrid vigour¹ or to prevent queen-only production due to the fixation of a caste-biasing genotype^{2,3}

Point # 2:

Once again, there is tension in the manuscript between levels of selection (individual versus colony). To me, the most exciting interpretation is that the evolution of xenoparity is a novel kind of major evolutionary transition in individuality (METI). This implication by itself, in my opinion, is enough to make it Nature worthy. Yet, the Authors give it only 1 sentence in the discussion. Furthermore, this interpretation of the data appears to be at odds with individual level explanations and associated use of terminology. For example, they frame the ancestral state as the queen “exploiting” another species’ sperm to make workers (implying using heterospecific sperm has benefits for worker production). But in paragraph beginning line 245 it is framed much more negatively (“locked into obligate sperm parasitism”, “best of a bad situation”). Of course, there are other loaded terms used, like “hijacking” (which I don’t like) and domestication, all of which are individual-level explanations that assign intention or agency to the individual workers reproductive organs, which don’t really make that much sense from the perspective of developmental biology or colony-level / systems evolution. From the superorganismal perspective, it would seem more reasonable to interpret the ancestral state as a mutualism not parasitism. What I mean is, there could be a mutual advantage, where hybrid workers are more fit and the second species benefits because its DNA is getting passed on (even if by another species). So I believe the authors should be a bit more open about the language / interpretations when reconstructing the evolutionary history and describe multiple possibilities except for just one. For example, it could have been any one of these evolutionary pathways:

Sperm parasitism → cross-species cloning / METI

Sperm mutualism → cross-species cloning / METI

Sperm parasitism → mutualism → cross-species cloning / METI

We share the enthusiasm of the referee about xenoparity as a novel kind of major evolutionary transition in individuality (METI). Following the suggestion, we have expanded the end of the discussion to better explicit this implication:

Major evolutionary transition in individuality occurs when distinct entities evolve into an integrated, higher-level unit⁴⁻⁶. As two species have become sexually interdependent into such an integrated entity, evolution toward xenoparity exemplifies how such transitions can occur via a sexual domestication process.

The referee correctly noted the use of terms with either negative or positive connotations regarding the ancestral system of *M. ibericus*, which stems from the ambiguous boundary between the evolutionary benefits and drawbacks of obligate parasitism. While a parasite initially exploits its host, it becomes locked in a dependency in case of obligate parasitism. To alleviate the fact that it may appear as at odds with the superorganismal perspective, we followed the referee’s suggestion and have replaced “hijacking”, “locked” or “best of bad situation” by more neutral terms. As suggested, we also mention that the evolutionary transition toward xenoparity may have been paved by various evolutionary steps along a parasitism-mutualism continuum. To illustrate this, we mention the example of *Messor barbarus*, which represents an alternative outcome from the ancestral state where the two partners are engaged in reciprocal sperm parasitism. This can be viewed as sperm mutualism at the superorganism level, as mentioned by the referee. Here are the resulting changes in the discussion when reconstructing the evolutionary history:

Transition toward xenoparity appears to result from sexual evolution along a parasitism-mutualism continuum. Like several other harvester ant species, *M. ibericus* first transitioned into obligate sperm parasitism^{1,7} (Fig. 3a), a situation where they lost the ability to produce workers by themselves due to epistatic incompatibilities^{2,8} or selfish caste-biasing genotypes³. **While not the most straightforward path towards xenoparity, this situation might have evolved towards reciprocal sperm parasitism, a form of sperm mutualism seen in other harvester ants where two lineages depend on each other’s sperm for worker production^{2,7,9}**

Point #3:

I appreciate the Authors providing further information (or at least as much as they could provide) about the mechanisms. This will be an interesting mechanism to crack in future studies, and I feel this would be worthy of saying in a single sentence at the end of the discussion. More importantly, however, I liked the figure they provided in the rebuttal more than the one they have in the paper and think they should consider switching it in.

Following the suggestion, we have added a sentence in the discussion about perspectives regarding the mechanism:

Investigating the male cloning mechanism will help to determine whether this developmental innovation is analogous to male parasitism¹⁰ or unique to the *M. ibericus* reproductive system.

We are pleased that the figure provided in the rebuttal has been well-received. We believe that the current Fig. 3 cannot be entirely replaced, as it is the only figure presenting morphological and genomic divergence between the clonal and wild-type lineages of *M. structor*. To ensure that the new figure will be readily available to all readers, we moved it from Supplementary information to Extended Data Fig. 7, so it will be directly accessible in the main PDF article.

Referee #2 (Remarks to the Author):

I reviewed an earlier version of this manuscript, which was subsequently rejected based on the comments of multiple referees. I have nevertheless been asked to review a resubmission of the manuscript, and to prevent delays in the decision I will keep my review brief. I have read the revised manuscript and the responses and by and large, my original review stands. The manuscript is very interesting and as far as I can assess the methodology, quite convincing. Its groundbreaking nature is a matter of interpretation and perspective, and while publication in a top journal is warranted, in my opinion the case for publication in Nature is marginal.

We value that our work has been assessed as both methodologically convincing and interesting. We acknowledge the subjective nature of evaluating impact and appreciate the reviewer's recognition that it deserves to be published in a top journal.

Referee #3 (Remarks to the Author):

Overall, the authors have done an excellent job in addressing reviewer comments and suggestions, and their effort is reflected in a more compelling and clear manuscript. I remain convinced that this article is of sufficient quality and general interest to warrant publication in Nature. I have several remaining wording suggestions- most are minor or are subjective comments, so I present them below in line number order:

We thank the reviewer to reaffirm that the article warrants publication in Nature and to have provided time for additional wording suggestions.

Line 26: should be "same-species OFFSPRING" (no s) – the word offspring can be singular or plural.

Done.

Line 28: lifecycle should be two words (life cycle)

We replaced all the occurrences of "lifecycle" by "life cycle".

Line 50: such a -> this

Done.

Lines 68-70: technically, I think that determining that the *M. ibericus* workers align with *M. ibericus* queens in the mitochondrial phylogeny confirms that they have mothers with *M. ibericus* mitogenomes, but doesn't rule out the possibility of some sort of clonal reproduction by workers. Without the other evidence provided later in the manuscript, this does not confirm that the *M. ibericus* queens mothered the workers. Rephrase this sentence to be more precise?

This is correct that this is a possibility at this stage of the manuscript, so we rephrased the sentence to talk about "*M. ibericus* maternal ancestry" instead of implying that *M. ibericus* queens necessarily mothered the workers one generation ago

The resulting tree suggests a *M. ibericus* maternal ancestry, as all hybrid workers share the mitochondrial genome of *M. ibericus* sexual individuals.

Lines 72-74: As above, this sentence "The resulting phylogenetic tree revealed that hybrid workers were all sired by *M. structor* males, as all paternal alleles (N=164) formed a well-supported clade with individuals of this species (Fig. S3)." does not rule out the possibility that workers reproduce clonally. I know that worker cloning would be unusual, but the reproductive system being reported is also strange. I suggest softening these statements until the additional lines of evidence supporting the xenoparity hypothesis are laid out.

Similarly, we now mention at this stage a "*M. structor* paternal ancestry" instead of stating that workers are all sired by *M. structor* males.

The resulting phylogenetic tree revealed that hybrid workers have a *M. structor* paternal ancestry, as all paternal alleles (N=164) formed a well-supported clade with individuals of this species

Lines 115-119 and supp info section 4: Given the presently available information about the clonal males, I still think that their species status is unclear. Perhaps defining the species concept that the authors are employing will help. Based on the biological species concept, we can't assess the species status (there is no evidence about whether the clonal males can or cannot cross with free-living *M. structor* females). Based on the morphological species concept, the clonal lineage is different from males of either species. The authors apparently rely on the phylogenetic species concept (since their tests all focus on genetic similarity of the nuclear genome). I can live with this assessment, though I note that there are numerous empirical examples of coadaptation between mitochondrial and nuclear genes (e.g. mitonuclear incompatibilities in hybridizing swordtails, <https://www.nature.com/articles/s41586-023-06895-8>); resulting behavioral or physiological differences based on the interaction between the mitochondrial and nuclear genome could affect the cohesiveness of a species. The additional nuclear analyses are still useful, but doubting the nuclear similarity of the two groups was not the basis of my initial comment. I appreciate the authors' additional efforts, but I still think that calling this these males by the name of the free-living species from which their nuclear genome was originally captured is premature. I also don't agree that the alternative (proposed in the response to reviews) is describing the clonal males as a new and distinct species at this stage (they are not capable of propagating themselves at any scale), but instead the hijacked genetic information that is propagated exclusively in *M. ibericus* colonies could be viewed as a portion of the essential genetic complement of *M. ibericus*. But as I said, as long as the authors are clear about how they are classifying the different branches of the *M. structor* species, I think this more semantic discussion can be delayed for now. The strange reproductive systems of ants will always evade some of the tenets of our classification system.

We thank the reviewer for sharing these relevant thoughts regarding species concepts and fully agree that the taxonomical status of clonal males is a matter of semantics, as they evade typical classification systems. As mentioned, we mostly rely on the phylogenetic species concept,

although the analysis about gene flow (Fig. S16b) tests historical reproductive isolation rather than genetic similarity, which is more in line with the biological species concept¹¹. We now mention that while clonal males are similar to their wild relatives regarding the phylogenetic species concept, they are distinct regarding the morphological species concept. We note that this ambiguity is shared by many domesticated lineages, which often exhibit divergent phenotypes in spite of high genetic similarity to their wild relatives. One more reason to qualify clonal males as a domesticated lineage is the parallel with the domestication of the mitochondrial genome within eukaryote cells (see Discussion), which closely aligns with the referee's view that clonal males could be viewed as hijacked genetic information now part of *M. ibericus*. We also appreciate the insightful comment regarding mitonuclear incompatibilities, and now recall that *M. structor* clonal males have *M. ibericus* mitochondria while citing the article raised by the referee. These points are now mentioned in the conclusion of Supplementary Note 4:

The concept of species has several definitions depending on the considered criteria, for example based on morphology (morphological species concept), genetic similarity (phylogenetic species concept) or reproductive isolation (biological species concept)^{11,12}. While it is clear that clonal males appear as different from wild-type *M. structor* according to the morphological species concept (Supplemental Text S3), they appear as part of *M. structor* according to the phylogenetic species concept (Fig. S14-16). The analysis about gene flow (Fig. S16b) tests reproductive isolation rather than genetic similarity, which suggests historical interfecundity between clonal and wild-type lineages, in line with the biological species concept.

While clonal males have extensively exchanged genes with the wild-type lineage through their history, they currently never or rarely hybridize with wild-type females (Fig. S13). This aligns with limited mating opportunities due to their recent "domesticated condition" within *M. ibericus* nests. As a domesticated lineage, they share the ambiguous taxonomic status of many domesticated species, which also often exhibit divergent phenotypes in spite of high genetic similarity to their wild relatives¹³. As they are produced by different species' mothers, domesticated *M. structor* males are also distinguished by a *M. ibericus* mitochondria, which may further limit mating with their wild relatives¹⁴.

Line 136: lived -> live?

Corrected

Lines 142-144: Out of curiosity, does this happen when producing conspecific males as well? In other words, have the authors investigated whether any of the *M. ibericus* males are not closely related to their mother? Not essential for this study, but it would be interesting to know more about the mechanism.

We did not retrieve evidence of male clonality for conspecific males, but the genetic diversity of *M. ibericus* is so low that it would be difficult to detect such a phenomenon in case of inbreeding. When investigating this question in one colony where we sequenced several individuals and their mother queen, we found that three out of three *M. ibericus* males (ORT3M1, ORT3M3 and ORT3M4) were identical to one of the allele of their mother (ORT3Q1). This suggests that, as far as we know, conspecific males are produced by unfertilized eggs, as is typically the case in Hymenoptera.

Referee #4 (Remarks to the Author):

This is really a wild discovery. I reviewed an earlier version of this manuscript. I have no further comments. Very exciting finding, the analyses are detailed, and the manuscript is well written.

We thank the reviewer for the positive assessment of our work for publication in Nature.

References

1. Umphrey, G. J. Sperm parasitism in ants: selection for interspecific mating and hybridization. *Ecology* **87**, 2148–2159 (2006).
2. Helms Cahan, S. & Keller, L. Complex hybrid origin of genetic caste determination in harvester ants. *Nature* **424**, 306–309 (2003).
3. Weyna, A., Romiguier, J. & Mullon, C. Hybridization enables the fixation of selfish queen genotypes in eusocial colonies. *Evol Lett* **5**, 582–594 (2021).
4. Maynard-Smith, J. & Szathmary, E. *The Major Transitions in Evolution*. vol. 49 (OUP Oxford, 1997).
5. West, S. A., Fisher, R. M., Gardner, A. & Kiers, E. T. Major evolutionary transitions in individuality. *Proc. Natl. Acad. Sci. U. S. A.* **112**, 10112–10119 (2015).
6. Rafiqi, A. M., Rajakumar, A. & Abouheif, E. Origin and elaboration of a major evolutionary transition in individuality. *Nature* **585**, 239–244 (2020).
7. Romiguier, J., Fournier, A., Yek, S. H. & Keller, L. Convergent evolution of social hybridogenesis in *Messor* harvester ants. *Mol. Ecol.* **26**, 1108–1117 (2017).
8. Anderson, K. E. *et al.* Distribution and evolution of genetic caste determination in *Pogonomyrmex* seed-harvester ants. *Ecology* **87**, 2171–2184 (2006).
9. Kuhn, A., Darras, H., Paknia, O. & Aron, S. Repeated evolution of queen parthenogenesis and social hybridogenesis in *Cataglyphis* desert ants. *Mol. Ecol.* **29**, 549–564 (2020).
10. Schwander, T. & Oldroyd, B. P. Androgenesis: where males hijack eggs to clone themselves. *Philos. Trans. R. Soc. Lond. B Biol. Sci.* **371**, (2016).
11. Stankowski, S. *et al.* Toward the integration of speciation research. *Evolutionary Journal of the Linnean Society* **3**, (2024).
12. De Queiroz, K. Species concepts and species delimitation. *Syst. Biol.* **56**, 879–886 (2007).
13. Zeller, U. & Göttert, T. The relations between evolution and domestication reconsidered - Implications for systematics, ecology, and nature conservation. *Global Ecology and Conservation* **20**, e00756 (2019).
14. Moran, B. M. *et al.* A lethal mitonuclear incompatibility in complex I of natural hybrids. *Nature* **626**, 119–127 (2024).

Referees' comments:

Referee #1 (Remarks to the Author):

The Authors have addressed all my comments from the previous round of review. As a result, the manuscript has much improved and is ready for publication in Nature.

We thank the referee for their help to improve the manuscript.